# On Imitation in Mean-field Games

Giorgia Ramponi[1]   Pavel Kolev[2]   Olivier Pietquin[3]   Niao He[4]
Mathieu Laurière[3,5]   Matthieu Geist[3]

[1] ETH AI Center, Zurich [2] Max Planck Institute for Intelligent Systems, Tübingen, Germany
[3] Google DeepMind [4] ETH Zurich, Department of Computer Science
[5] Shanghai Frontiers Science Center of Artificial Intelligence and Deep Learning, NYU Shanghai

## Abstract

We explore the problem of imitation learning (IL) in the context of mean-field games (MFGs), where the goal is to imitate the behavior of a population of agents following a Nash equilibrium policy according to some unknown payoff function. IL in MFGs presents new challenges compared to single-agent IL, particularly when both the reward function and the transition kernel depend on the population distribution. In this paper, departing from the existing literature on IL for MFGs, we introduce a new solution concept called the Nash imitation gap. Then we show that when only the reward depends on the population distribution, IL in MFGs can be reduced to single-agent IL with similar guarantees. However, when the dynamics is population-dependent, we provide a novel upper-bound that suggests IL is harder in this setting. To address this issue, we propose a new adversarial formulation where the reinforcement learning problem is replaced by a mean-field control (MFC) problem, suggesting progress in IL within MFGs may have to build upon MFC.

## 1   Introduction

Imitation learning (IL) is a popular framework involving an apprentice agent who learns to imitate the behavior of an expert agent by observing its actions and transitions. In the context of mean-field games (MFGs), IL is used to learn a policy that imitates the behavior of a population of infinitely-many expert agents that are following a Nash equilibrium policy, according to some unknown payoff function. Mean-field games are an approximation introduced to simplify the analysis of games with a large (but finite) number of identical players, where we can look at the interaction between a representative infinitesimal player and a term capturing the population's behavior. The MFG framework enables to scale to an infinite number of agents, where both the reward and the transition are population-dependent. The aim is to learn effective policies that can effectively learn and imitate the behavior of a large population of agents, which is a crucial problem in many real-world applications, such as traffic management [12, 30, 31], crowd control [11, 1], and financial markets [6, 5].

IL in MFGs presents new challenges compared to single-agent IL, as both the (unknown) reward function and the transition kernel can depend on the population distribution. Furthermore, algorithms will depend on whether we can only observe the trajectories drawn from the Nash Equilibrium (NE) or if we can access the MFG itself, either driven by the expert population or the imitating one.

The main question we address is whether IL in MFGs is actually harder than IL in single-agent settings and if we can use single-agent techniques to solve IL in the MFGs framework.

Although there exist IL algorithms for MFGs in the literature, none comes with a characterization of the quality of the learnt imitation policy. We will also explain that they essentially amount to a reduction to classic IL, and explain the underlying possible issues. So as to compare algorithms on a

37th Conference on Neural Information Processing Systems (NeurIPS 2023).

rational basis, we provide an extension of the concept of imitation gap to this setting and study it. Our contributions are:

- We introduce a new solution concept for IL in MFGs called *Nash imitation gap*, which is a strict generalization of the classic imitation gap and that we think may be more widely applicable to Multi-agent Reinforcement Learning.

- In light of this new criterion, we first study the setting where only the reward depends on the population's distribution, while the dynamics does not. This setting was largely studied in the past few years [16, 4, 1, 14, 22], and we show that in this case IL in MFGs reduces to single-agent IL with similar guarantees for Behavioral Cloning (BC) and Adversarial Imitation (ADV) type of algorithms.

- Then, we provide a similar analysis in the more general setting where the dynamics depends on the population's distribution. In this case, we provide for BC and ADV approaches upper-bounds that are exponential in the horizon, suggesting that IL is harder in this setting. On an abstract way, all previous works of the existing literature correspond to this setting.

- Due to these negative results, we introduce a new proxy to the Nash imitation gap, for which we can derive a quadratic upper bound on the horizon. Then, we discuss how a practical algorithm could be designed with an adversarial learning viewpoint. The idea behind it is to use an approach similar to adversarial IL, where the underlying RL problem is replaced by a Mean-Field Control (MFC) problem. We leave the design and experimentation of practical algorithms for future works, but this suggests that making progress on IL in MFGs may have to build upon MFC. We also provide a numerical illustration empirically supporting our claims in the appendix.

## 2 Background

### 2.1 Mean-field Games

Intuitively, an MFG corresponds to the limit of an $N$-player game when $N$ tends towards infinity. We focus on the finite-horizon setting, in discrete time, and with finite state and action spaces. Mathematically, the MFG is defined by a tuple $\mathcal{M} = (\mathcal{S}, \mathcal{A}, P, r, H, \rho_0)$ where $\mathcal{S}$ is a finite state space, $\mathcal{A}$ is a finite action space, $P : \mathcal{S} \times \mathcal{A} \times \Delta_{\mathcal{S}} \to \Delta_{\mathcal{S}}$ is a transition kernel, $r : \mathcal{S} \times \mathcal{A} \times \Delta_{\mathcal{S}} \to \mathbb{R}$ is a reward function, $H$ is a finite horizon and $\rho_0 \in \Delta_{\mathcal{S}}$ is a distribution over initial states. The first and second inputs of $P$ and $r$ represent respectively the individual player's state and action, while the third input represents the distribution of the population. This allows the model to capture interactions or mean field type. We denote $[H - 1] = \{0, \ldots, H - 1\}$. Since the problem is set in finite time horizon, we consider non-stationary stochastic policies of the form $\pi = (\pi_0, \ldots, \pi_{H-1})$ where for each $n \in [H - 1]$, $\pi_n : \mathcal{S} \to \Delta_{\mathcal{A}}$. This means that, at time $n$, a representative player whose state is $s_n \in \mathcal{S}$ picks an action according to $\pi_n(s_n)$. We will also view $\pi_n$ as a function from $\mathcal{S} \times \mathcal{A}$ to $[0, 1]$ and use the notation $\pi_n(a|s_n)$ to denote the probability to pick action $a$ according to $\pi_n(s_n)$. We also introduce the set of population-independent non-stationary rewards uniformly bound by 1, that will be useful later, $\mathcal{R} = \{r : \mathcal{S} \times \mathcal{A} \times [H - 1] \to [-1, 1]\}$.

When a player is at state $s$ and uses action $a$ while the population distribution is $\rho$, it gets the reward $r(s, a, \rho)$. In a finite-player game, $\rho$ would be replaced by the empirical distribution of the other players' states. These players would be influenced by the player under consideration, leading to complex interactions. However, the influence of a single player on the population distribution becomes smaller as the number of players increases. In the limit, we can expect that each player has no influence on the distribution. We will use interchangeably the terms "agent" and "player".

The MFG framework allows us to formalize this idea. So the problem faced by a single representative player is a Markov Decision Process (MDP), in which the distribution is fixed: Given a mean field sequence $\rho = (\rho_n)_{n \in [H-1]}$, the player wants to find a policy $\pi$ maximizing the value function $V$ defined by:

$$V(\pi, \rho) = \mathbb{E}\left[\sum_{n=0}^{H-1} r(S_n, A_n, \rho_n)\right]$$

where $S_0$ is distributed according to the initial state distribution $\rho_0$, $A_n \sim \pi_n(S_n)$, and $S_{n+1} \sim P(S_n, A_n, \rho_n)$. The resulting policy is called a best response to the mean field $\rho$. We will sometime write $V_r(\pi, \rho)$ to make explicit to reward of the value function.

**Definition 1** (Best response). *A policy $\pi$ is a best response to $\rho$ if: $V(\pi, \rho) = \max_{\pi'} V(\pi', \rho)$.*

Intuitively, $\rho$ is a Nash equilibrium if it is the distribution sequence obtained when all the players use a best response policy against $\rho$. To formalize this idea, we introduce the notion of population distribution sequence induced by a policy that does influence the transition kernel.

**Definition 2** (Population distribution sequence). *We denote by $\rho^{(\pi)}$ the state-distribution sequence induced by the population following policy $\pi$, which is defined as:*

$$
\begin{cases}
\rho_0^{(\pi)}(s) & = & \rho_0(s), & s \in \mathcal{S} \\
\rho_{n+1}^{(\pi)}(s') & = & \sum_s \rho_n^{(\pi)}(s) \sum_a \pi_n(a|s) P(s'|s, a, \rho_n^{(\pi)}), & (n, s') \in [H-1] \times \mathcal{S}.
\end{cases}
$$

*The state-action distribution sequence, denoted by $\mu^{(\pi)}$, is defined as:*

$$
\mu_n^{(\pi)}(s, a) = \pi_n(a|s) \rho_n^{(\pi)}(s), \qquad (n, s, a) \in [H-1] \times \mathcal{S} \times \mathcal{A}.
$$

We can now give a formal definition of Nash equilibrium.[1]

**Definition 3** (Nash equilibrium). *A policy $\pi$ is called a Nash equilibrium policy if it is a best response against $\rho^\pi$, i.e., $\pi$ is a maximizer of $\pi' \mapsto V(\pi', \rho^\pi)$. The distribution sequence $\rho^\pi$ induced by a Nash equilibrium policy $\pi$ is called a Nash equilibrium mean field sequence.*

The value function can be written without expectation by introducing the state and state-action distributions for a single agent, which does not influence the transition kernel (only the population does, it is a core reason for considering the mean-field limit).

**Definition 4** (Single-agent distribution sequence). *Consider an agent using policy $\pi'$ who evolves among a population using policy $\pi$. We denote by $\rho^{(\pi)\pi'}$ the state-distribution sequence of this single agent, which is defined by:*

$$
\begin{cases}
\rho_0^{(\pi)\pi'}(s) & = & \rho_0(s), & s \in \mathcal{S} \\
\rho_{n+1}^{(\pi)\pi'}(s') & = & \sum_s \rho_n^{(\pi)\pi'}(s) \sum_a \pi'_n(a|s) P(s'|s, a, \rho_n^{(\pi)}), & (n, s') \in [H-1] \times \mathcal{S}.
\end{cases}
$$

*The state-action distribution sequence is defined as:*

$$
\mu_n^{(\pi)\pi'}(s, a) = \pi'_n(a|s) \rho_n^{(\pi)\pi'}(s), \qquad (n, s, a) \in [H-1] \times \mathcal{S} \times \mathcal{A}.
$$

From these definitions we directly have that $\rho^{(\pi)\pi} = \rho^{(\pi)}$ and $\mu^{(\pi)\pi} = \mu^{(\pi)}$. Furthermore, for any policies $\pi$ and $\pi'$, we define a generalized value function

$$
V(\pi', \rho^{(\pi)}) = \sum_{n=0}^{H-1} \sum_{s,a} \mu_n^{(\pi)\pi'}(s, a) r(s, a, \rho_n^{(\pi)}).
$$

**Definition 5** (Exploitability). *The exploitability of a policy $\pi$ quantifies the gain for a representative player to replace its policy by a best response:*

$$
\mathcal{E}(\pi) = \max_{\pi'} V(\pi', \rho^{(\pi)}) - V(\pi, \rho^{(\pi)}).
$$

A Nash equilibrium policy can be defined equivalently as a policy such that its exploitability is $0$.

Finding a Nash Equilibrium policy is different from trying to maximize the value function, which can be interpreted as a social optimum. This problem is sometimes referred to as mean field control problem (MFC) because it can be interpreted as an optimal control for an MDP where the state is augmented with the distribution.

**Definition 6** (Social optimum). *A policy $\pi$ is socially optimal if $V(\pi, \rho^\pi) = \max_{\pi'} V(\pi', \rho^{\pi'})$.*

In general the Nash equilibrium and the social optimum do not coincide, i.e., the MFG policy and the MFC solution can be different. In other words, if $\pi$ is an MFG solution, then we might have $V(\pi, \rho^\pi) < \max_{\pi'} V(\pi', \rho^{\pi'})$, and if $\pi$ is an MFC policy, then we might have $\mathcal{E}(\pi) \neq 0$.

In this work, we will frequently make two common assumptions (e.g., [2, Asm. 1], or [34, Asm. 1]).

---

[1] We will sometimes omit the term "Nash" and simply write "equilibrium" when the context is clear.

**Assumption 1.** *The reward function $r$ and the transition kernel $P$ are Lipschitz continuous w.r.t. the population distribution and have corresponding Lipschitz constants $L_r$ and $L_P$. In particular, for any state-action pair $(s, a)$ it holds for any state distributions $\rho, \rho' \in \Delta_S$ that*

$$|r(s, a, \rho) - r(s, a, \rho')| \leq L_r \|\rho - \rho'\|_1.$$
$$\|P(\cdot|s, a, \rho) - P(\cdot|s, a, \rho')\|_1 \leq L_P \|\rho - \rho'\|_1.$$

These assumptions mean that the reward and the transitions depend on the mean field in a smooth way. For example, $r(s, a, \rho) = -c_r \rho(s) + \tilde{r}(s, a)$ satisfies the assumption with $L_r = c_r$. A reward of this form penalizes the agent for being in a crowded state. In particular, these assumptions imply that $r$ and $P$ are continuous with respect to the distribution and since $S$ and $A$ are finite sets and we consider a discrete-time, finite horizon MFG, these assumptions are sufficient to ensure existence of a Nash equilibrium, see e.g. [10, Proposition 1].

Note that these assumptions do not assume that $L_r$ or $L_P$ need to be small. However, we will consider separately the case $L_P = 0$, which has received attention in the MFG literature and which corresponds to situations where the mean-field interactions occur through the reward only. Note that when $L_P = L_r = 0$, then there are no interactions so the Nash equilibrium condition becomes trivial and the problem reduces to a single agent MDP. We will write the value function as $V(\pi)$.

## 2.2 Classic imitation learning

The single-agent setting is a special case of the above, with $L_P = L_r = 0$. In this case, the Imitation Learning (IL) problem has been extensively studied [20]. In IL we assume that we observe state-action trajectories generated by an expert who is using an optimal policy $\pi^E$, i.e., $\pi^E \in \arg\max V(\pi)$. We do not know $\pi^E$, $V$, $r$ nor $P$. The goal is to learn a policy, denoted by $\pi^A$, which performs as well as the expert policy $\pi^E$ according to the unknown reward: $V(\pi^A) = V(\pi^E) = \max_{\pi'} V(\pi')$.

Given the imitation policy $\pi^A$, the *imitation gap* is a non-negative quantity defined by $V(\pi^E) - V(\pi^A)$ [29]. The goal is to learn a policy $\pi^A$ whose imitation gap is as close to $0$ as possible. However, when we do not know the model, this quantity cannot be minimized directly. For this reason, different methods were introduced in literature. Here we focus on the following two prominent methods: Behavioral Cloning (BC) [24] and adversarial IL such as Generative Adversarial IL (GAIL) [18].

For simplicity, we will denote $\rho^E = \rho^{(\pi^E)\pi^E}$, $\mu^E = \mu^{(\pi^E)\pi^E}$, $\rho^A = \rho^{(\pi^A)\pi^A}$, and $\mu^A = \mu^{(\pi^A)\pi^A}$. When the dynamics does not depend on the population ($L_P = 0$), we will write $\rho^\pi = \rho^{(\pi^{\text{any}})\pi}$, $\mu^\pi = \mu^{(\pi^{\text{any}})\pi}$, as the population driven by $\pi^{\text{any}}$ has no effect on the transition kernel.

**Behavioral Cloning.** In this work, we frame BC as minimizing the expected $\ell_1$-distance between the action probability distributions of the expert and the imitation policy, where the expectation is over the expert state occupancy. Although in practice it consists in a reverse KL-divergence, or equivalently as the maximum likelihood estimation in supervised learning, here we consider the $\ell_1$-norm distance (or scaled total variation), which is more convenient for the analysis. However, the analysis could be adapted to this maximum likelihood estimation with minor changes thanks to Pinsker's inequality (see App. A). In the finite-horizon case, one obtains a bound on quantities of the form:

$$\epsilon_n^{\text{BC}} := \mathbb{E}_{s \sim \rho_n^E}[\|\pi_n^A(s) - \pi_n^E(s)\|_1], \qquad n \in \{0, \dots, H-1\}. \tag{1}$$

That is, we solve one BC problem per time-step, from data generated by an expert. The single-agent imitation-learning gap is bounded by $\mathcal{O}\left(\epsilon^{\text{BC}} H^2\right)$, where $\epsilon^{\text{BC}} = \max_{n \in \{0, \dots, H-1\}} \epsilon_n^{\text{BC}}$ [26]. We will retrieve this result as a special case of our analysis.

**Adversarial Imitation Learning.** Using adversarial-like approaches [18, 13] we control a distance or divergence between the state-action occupancy measures of the expert and the imitation policy. In the single-agent IL setting this divergence can be expressed as:

$$\epsilon_n^{\text{ADV}} := \|\mu_n^E - \mu_n^A\|_1, \qquad n \in \{0, \dots, H-1\}. \tag{2}$$

Similar IL quantities are extensively studied in the single-agent case, leading to algorithms minimizing a divergence or distance such as Generative Adversarial Imitation Learning [18] or IQ-learn [13]. If we control the state-action occupancy measure errors, then we get a single-agent imitation-learning gap of order $\mathcal{O}\left(\epsilon^{\text{ADV}} H\right)$, where $\epsilon^{\text{ADV}} = \max_{n \in [0, \dots, H-1]} \epsilon_n^{\text{ADV}}$, gaining an $H$ factor compared to the BC bound [32].

To make the link to adversarial approaches, controlling the terms of Eq. (2) can be achieved through the following equivalent formulation, based on the Integral Probability Metric (IPM) formulation of the total variation (see Appx. B for a detailed derivation and additional discussions):

$$\min_{\pi} \sum_{n=0}^{H-1} \|\mu_n^E - \mu_n^\pi\|_1 = \max_{f \in \mathcal{R}} \min_{\pi} (V_f(\pi^E) - V_f(\pi)), \tag{3}$$

with $\mathcal{R}$ defined in Sec. 2.1 (non-stationary population-independent rewards functions uniformly bounded by 1). This form is similar to classic adversarial imitation approach, learning both a reward function and a policy, the inner problem being a reinforcement learning (RL) problem. Previous work usually consider an $f$-divergence [15, 19] between state-action occupation measures, which leads to different min-max problems. However, in this paper, we will focus on the $\ell_1$-distance, that we think is a meaningful and practical abstraction of adversarial approaches. Our results could be extended to other settings by using tools like the Pinsker inequality.

**Remark 1.** *Remember that in the single-agent case (same as when we have $L_P = L_r = 0$) the transition dynamics and the reward function do not depend on the population distribution. In the mean-field game, on the other hand, the transition dynamics and the reward function can depend on the population distribution. For this reason, we need to consider different quantities to be controlled. We will see more about it in the next section.*

## 3  Related works

We provide a commented review of the literature focusing on the fundamentals of the different approaches, their pros and their cons.

To the best of our knowledge, the first work addressing the IL problem in MFGs is [33]. The authors consider a discrete-time MFG over a complete graph, and they propose a reduction from MFGs to finite-horizon MDP (with a population-augmented state) and then use single-agent IL algorithms on the new MDP. In the reduction, the new reward function is computed in the following way: $\overline{r}(\rho, \pi) = \sum_s \rho(s) \sum_a \pi(a|s) r(s, a, \rho)$, considering that the state is the population distribution $\rho$ and the actions are the possible policies $\pi$. Using this reduction, the authors implicitly assume that the observed expert is solving an MFC problem, i.e. she is looking for a socially optimal policy (see Def. 6). However, in general, this does not actually coincide with a Nash Equilibrium policy (see Def. 3). Then, the reduction works only for cooperative MFGs, but it is prone to biased reward inference in non-cooperative environments.

The first work enlightening this issue is [8]. The authors consider a discounted finite-horizon MFG and propose a novel method called Mean Field IRL. They reframe the problem as finding a reward function that makes the expert policy $\pi^E$ the best response with respect to the expert population distribution $\rho^E$, i.e., find a population-dependent $r$ such that $V(\pi^E, \rho^E) \in \text{argmax}_\pi V(\pi, \rho^E)$. To solve this problem they use a max-margin approach similar to [25] in the single-agent case. In fact, fixing the population distribution $\rho^E$, the MFG is reduced to an MDP, and the imitation problem is reduced to single-agent IL. However, since they do not have access to the actual population distribution $\rho^E$, they need to estimate it from samples. In this way, the authors do not have any actual theoretical guarantee on the performances of the recovered policy, since it depends on the estimation of $\rho^E$.

Recently, [9] proposed a novel approach reusing ideas from adversarial learning [18] and maximum entropy [36]. The authors assume they have access to an expert optimizing a mean-field-regularized Nash-equilibrium, i.e., the value-function has an additional regularization term: $V(\pi', \rho^\pi) = \sum_{n=0}^{H-1} \sum_s \left\{ \mathcal{H}(\pi'(\cdot|s)) + \sum_a \mu_n^{(\pi)\pi'}(s, a) r(s, a, \rho_n^{(\pi)}) \right\}$, with $\mathcal{H}$ the Shannon entropy. This term guarantees that the MFG has a unique equilibrium solution, and only one best response policy. Then, they assume to have access to the population distribution $\rho^E$, or to be able to estimate this population distribution from samples. Fixing the population distribution, the MFG problem reduces to the single-agent setting, which allows applying GAIL [18] or maximum entropy IRL to learn an approximate policy $\pi^A$. This leads to two main problems. (i) We are actually estimating $\rho^E$ from samples, trying to find a policy which gives us the true $\rho^E$; this circulating reasoning leads to many estimation errors. (ii) From an abstract viewpoint, the final theoretical guarantee we can hope to have for the recovered policy $\pi^A$ would approximately be $\|\mu^{(\pi^E)\pi^A} - \mu^{(\pi^E)\pi^E}\|_1$ (in practice for a different distance or divergence, but involving the same occupancy measures), which may not

be sufficient as we will show in Sec. 4.2. We will see that the upper-bound for this case has an exponential dependency on the horizon.

In [35], the authors assume observing an MFG where the agents are acting according to a correlated equilibrium. The authors justify the study of a correlated equilibrium by some applications where we can have access to some correlation device (e.g., traffic network equilibrium induced from the public routing recommendations). Similar to [9], the authors of this work reuse ideas from adversarial learning [18] to recover a policy observing a correlated equilibrium policy. As in [9] they fix the population distribution to be the expert one and then apply single-agent GAIL to the problem. Then, although the solution concept is different, this approach faces similar problems as in [9], where instead the authors considered to observe an expert following a Nash Equilibrium policy. We will focus on experts achieving a Nash equilibrium. Extending our results (see Sec. 4) to more general equilibria such as coarse correlated ones is an interesting future research direction.

Notice also that all GAIL-like approaches discussed above learn an intermediate population-dependent reward function. This is complicated (as the population is a distribution over the state space, generally difficult to represent compactly), and in fact superfluous in their setting. Indeed, as all these approaches assume that the interaction is done with the MFG driven by the expert population (or a given approximation of it), it is sufficient to consider an intermediate population-independent but non-stationary reward. This will appears clearly in the IPM formulations we propose in Sec. 4.

## 4 Nash imitation gap and imitation in MFGs

We consider the imitation learning problem in MFGs. Similar to single-agent IL, we observe the interactions between an expert and a fixed MFG environment, for which we do not know the reward function $r$ nor the transition kernel $P$. We only observe samples coming from an expert policy denoted by $\pi^E$, which we assume to be a Nash equilibrium policy, i.e., $\mathcal{E}(\pi^E) = 0$. In single-agent IL the goal is clear: Find a policy $\pi^A$ to minimize the imitation gap $\max_\pi V(\pi) - V(\pi^A)$. However, in MFG, the goal is less clear. Previous works focused on solving IL for a single agent in a population that stays at equilibrium, but this is not relevant for many applications, in which the population might use the learnt policy. The learnt policy should thus not only be good at the individual level, but it should also be an equilibrium policy. As a first contribution, we propose a natural formulation for studying the performance of imitation learning in MFG, called Nash imitation gap:

**Definition 7.** *The Nash imitation gap (NIG) of a policy $\pi^A$ is defined as:*
$$\mathcal{E}(\pi^A) = \max_{\pi' \in \Pi} V(\pi', \rho^{(\pi^A)}) - V(\pi^A, \rho^{(\pi^A)}).$$

Therefore, the NIG is simply defined as being the exploitability of the considered policy. The NIG has many interesting and useful properties. (i) If it is zero, the recovered policy $\pi^A$ is a Nash equilibrium policy. (ii) It is a generalization of the single-agent imitation gap (see above and Sec. 2.2), which is recovered as a special case when $L_P = L_r = 0$.

As for the single-agent imitation gap, we cannot optimize it directly, since we do not know the reward function, but instead we can envision proxys, such as reducing the distance between the recovered policy $\pi^A$ and the expert policy $\pi^E$ (BC-like) or their occupancy measures (GAIL-like) as in the classic IL setting (see Sec. 2.2). In this section, we first discuss the imitation learning problem when the dynamics does not depend on the population, i.e. $L_P = 0$, a common setting largely explored in the last years [22, 23]. Then, we present our results for the general setting when the dynamics depends on the population, i.e. when $L_P > 0$.

For what follows, in addition to the Lipschitz assumption (Asm. 1), we will also assume that the unknown reward function $r$ for which the expert $\pi^E$ is a Nash equilibrium is uniformly bounded when the population is the expert one.

**Assumption 2.** *The unknown reward $r$ satisfies*
$$\max_{s,a} |r(s, a, \rho^{(\pi^E)})| \leq r_{max}.$$

### 4.1 Population-independent dynamics: a reduction to classic imitation

We start by analyzing a simpler but commonly used setting (e.g., see [23] in the context of reinforcement learning methods, or [16, 4, 1] in the context of the analysis of discrete or continuous space

MFGs), where the dynamics does not depend on the population. Here the MFG interaction is without reward and it is only the (unknown) reward function that depends on the population. This setting is equivalent to observing an interaction between an expert and an MDP, since the state-distribution does not depend on the population, i.e., $\rho^{(\pi)\pi'} = \rho^{\pi'}$.

**Behavioral Cloning.** The behavioral cloning setting is the same as the single agent setting (see Sec. 2.2), i.e, we control:

$$\epsilon_n^{\mathrm{BC}} := \mathbb{E}_{s \sim \rho_n^E}[\|\pi_n^A(s) - \pi_n^E(s)\|_1], \qquad n \in \{0, \ldots, H-1\}, \tag{4}$$

where $\pi^E$ is the expert policy and $\pi^A$ the imitation policy. Under the assumption of the BC-type error, we give the following bound on the Nash imitation gap (proof in Appx. C).

**Theorem 1.** *Let $\epsilon^{\mathrm{BC}} = \max_{n \in \{0, \ldots, H-1\}} \epsilon_n^{\mathrm{BC}}$. If $L_P = 0$, the Nash imitation gap satisfies*

$$\mathcal{E}(\pi^A) \leq H^2(2L_r + r_{max})\epsilon^{\mathrm{BC}}.$$

Theorem 1 shows that when $L_P = 0$ we recover the single-agent imitation learning bound. Perhaps surprisingly, the simple BC approach was not previously considered for MFG, to the best of our knowledge.

**Adversarial learning.** In the adversarial setting, similar to the single-agent case (see Sec. 2.2) we control a distance or divergence between occupancy measures. More precisely, we consider the following distance between occupancy measures:

$$\epsilon_n^{\mathrm{ADV}} := \|\mu_n^{\pi^A} - \mu_n^{\pi^E}\|_1, \qquad n \in \{0, \ldots, H-1\}.$$

It is important to recall that in this setting, the dynamics do not depend on the population, thus $\mu^{(\pi)\pi''} = \mu^{(\pi')\pi''}$ for every triplet of policies $(\pi, \pi', \pi'')$.

Since in this case the dynamics do not depend on the population distribution, the same IPM approach (see Eq. (3)) of single-agent imitation-learning also works in this context. Then we can practically solve the MFG IL problem using GAIL-like approaches [18, 15, 19, 13].

We now provide a novel bound on the Nash imitation gap (proof in Appx. C).

**Theorem 2.** *Let $\epsilon^{\mathrm{ADV}} = \max_{n \in \{0, \ldots, H-1\}} \epsilon_n^{\mathrm{ADV}}$. If $L_P = 0$, the Nash imitation gap satisfies*

$$\mathcal{E}(\pi^A) \leq H(2L_r + r_{max})\epsilon^{\mathrm{ADV}}.$$

In contrast to the quadratic horizon dependence in Theorem 1, we derive here a linear horizon dependence. Furthermore, the bound in Theorem 2 is almost the same (similarly to the BC case) as in the single-agent IL problem. In fact, it recovers the bound of [32] by setting $L_r = 0$.

**Discussion.** When $L_P = 0$ interacting with the MFG without reward amounts to interact with and MDP without reward, so from a practical aspect any classic IL approach could be used, including Dagger-like approaches [27] that we do not analyse here. Our upper bounds show that when the dynamics is independent of the population, IL for MFG has similar guarantees as in single-agent imitation learning. In fact, our results suggest that a population-dependent unkown reward function affects the IL policy performance very moderately (through the $L_r$ constant). However, new challenges may arise if we also want to recover the reward function, as in the case of Inverse Reinforcement Learning. We leave this interesting research direction for future work.

## 4.2 Population-dependent dynamics

When the dynamics of the MFG depend on the population, i.e., $L_P > 0$, then the previous results do not apply anymore. In this section, we present results on MFGs with population-dependent dynamics, for the same proxys introduced above (BC and adversarial).

**Behavioral Cloning.** The BC proxy to the NIG is the same as in Equation (4). In this case, however, the bound we get is no longer comparable to the classic IL setting (proof in Appx. C).

**Theorem 3.** *Let $\epsilon^{\mathrm{BC}} = \max_{n \in \{0, \ldots, H-1\}} \epsilon_n^{\mathrm{BC}}$. If $L_P > 0$, the Nash imitation gap satisfies:*

$$\mathcal{E}(\pi^A) \leq \left(\frac{2(L_r + r_{max})}{L_P^2}(1 + L_P)^H + r_{max}H^2\right)\epsilon^{\mathrm{BC}}.$$

We observe that when the dynamics depend on the population, the dependence on the horizon is no longer quadratic but exponential. From a technical viewpoint, this comes from the dependency of the transition kernel to the population. A worse dependency makes sense intuitively. In the classic IL setting, an error on policies will amplify at the occupancy measure level (drift phenomenon in imitation learning). Here, this problem is even more amplified by the fact that the transition kernel that defines the occupancy measure itself depends on the related population, amplifying even more the imitation error. Sec. 5 provides some empirical evidence that IL in MFGs has an exponential dependence on the horizon for BC as $L_P$ increases.

Although we cannot claim our result is tight, this suggests that in MFGs we cannot hope to use BC to obtain a good imitation policy $\pi^A$, but we need to control the divergence between the state-action occupancy induced by $\pi^A$ and the one induced by $\pi^E$.

**Adversarial learning.** In contrast to the population-independent case, the quantity to control in this setting is not trivial. In fact, it depends on the MFG interaction assumption. We can assume to have the possibility to interact with only the MFG or with the MFG driven by $\rho^E$. These two types of interactions lead us to consider two different errors. We start by considering the error we would like to minimize if we can interact with the MFG driven by the expert population $\rho^E$. Assuming access to this MFG, the IL problem reduces to doing classic IL in an MDP without reward and with transition distribution $P(\cdot|s, a, \rho^{\pi^E})$. The error is defined as follows:

$$\epsilon_n^{\text{vanilla-ADV}} := \|\mu_n^{(\pi^E)\pi^A} - \mu_n^{(\pi^E)\pi^E}\|_1, \quad n \in \{0, \dots, H-1\}.$$

This assumption was implicitly made in all previous works (see Sec. 3), where the authors assume to fix the expert distribution $\rho^E$ and then solve the IL problem. However, assuming to have access to the expert distribution may not be reasonable in practice, and due to this in [9, 8] the authors replace the expert population distribution $\rho^E$ with its approximation from sampling $\hat{\rho}^E$, in order to learn $\mu^{(\pi^E)\pi^A}$. In reality, however, we seek to learn $\mu^{(\pi^E)\pi^A}$ and then $\rho^E$, which is the ultimate goal of the problem. This circular reasoning leads to not easily having theoretical guarantees for this setting.

In this case, for obtaining an adversarial formulation, we can use a similar approach as the one for the non-population dependent dynamics, Eq. (3), providing (details in Appx. B):

$$\min_\pi \sum_{n=0}^{H-1} \|\mu_n^{(\pi^E)\pi^E} - \mu_n^{(\pi^E)\pi}\|_1 = \max_{f \in \mathcal{R}} \min_\pi (V_f(\pi^E, \rho^{(\pi^E)}) - V_f(\pi, \rho^{(\pi^E)})).$$

We can observe that the inner problem is again an RL problem (for the MDP induced by $\rho^{(\pi^E)}$), and in practice any single-agent adversarial approach could be applied to solve for a similar proxy (related to a different min-max problem).

We provide a bound for this approach (proof in Appx. C).

**Theorem 4.** *Let* $\epsilon^{\text{vanilla-ADV}} = \max_{n \in \{0, \dots, H-1\}} \epsilon_n^{\text{vanilla-ADV}}$. *If* $L_P > 0$ *the NE imitation gap satisfies:*

$$\mathcal{E}(\pi^A) \leq \left(\frac{2(L_r + r_{\max})}{L_P}(1 + L_P)^H + r_{\max}H\right)\epsilon^{\text{vanilla-ADV}}.$$

Therefore, in this case, if in practice we can apply GAIL-like approaches to the MDP with transition distribution $P(\cdot|s, a, \rho^E)$, we have an exponential dependency in the horizon. Interestingly this is close to the quantity that previous works [9, 8] tried to control. Although we do not know if this bound is tight, reasonably controlling this quantity can lead to weak theoretical guarantees. Sec. 5 provides empirical evidences that, when $H$ and $L_P$ are large enough, vanilla-ADV exponential depends on the horizon like BC.

### 4.3  Population-dependent dynamics: a new efficient proxy

Although the vanilla-ADV error is a reasonable quantity to control, another interesting proxy is to consider state-action occupancy measures with population induced by the associated policy, namely

$$\epsilon_n^{\text{MFC-ADV}} := \|\mu_n^{(\pi^A)\pi^A} - \mu_n^{(\pi^E)\pi^E}\|_1, \quad n \in \{0, \dots, H-1\}.$$

To control this quantity, we do not need to have access to the expert population (except through data), we can directly interact with the MFG, driven by what we are learning. As we explained in Sec. 4.1,

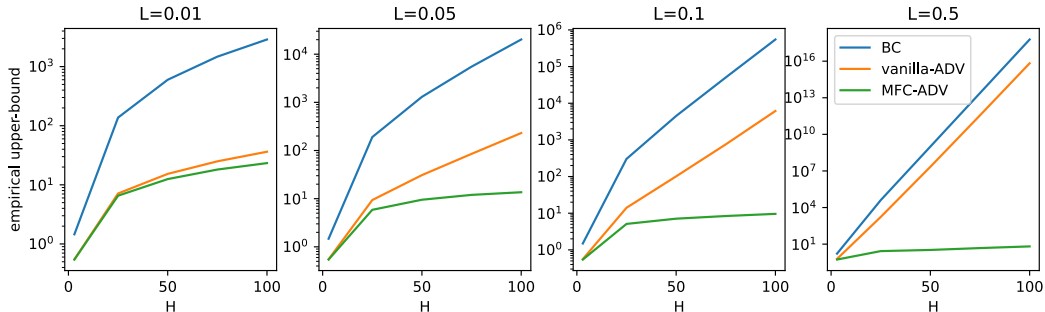

Figure 1: For each of the curves in Fig. 1, each one for a specific value of $L$, of $H$, and for a type of error (BC, vanilla-ADV and MFC-ADV), we consider the points $(\text{NIG}_k, \epsilon_k)$ making the curve, and compute the value $\max_k\left(\frac{\text{NIG}_k}{\epsilon_k}\right)$ as an empirical upper-bound for the specific value of $L$, $H$ and kind of error. Then, for each value of $L$ and each kind of error, we plot this empirical upper-bound as a function of the horizon $H$, with a log-scale on the $y$-axis.

if $L_P = 0$ then this quantity is the same as the vanilla-ADV error. Before presenting and discussing an adversarial formulation (that will explain the naming choice for the error), we provide a novel bound for this quantity (proof in Appx. C):

**Theorem 5.** *Let $\epsilon^{\text{MFC}-\text{ADV}} = \max_{n\in\{0,...,H-1\}} \epsilon_n^{\text{MFC}-\text{ADV}}$. If $L_P > 0$, the Nash imitation gap satisfies:*

$$\mathcal{E}(\pi^A) \leq \left[3L_P r_{\max} H^2 + (2L_r + r_{\max})H\right] \epsilon^{\text{MFC}-\text{ADV}}.$$

This gives us a significant improvement compared to the BC case and the vanilla-ADV error. Indeed, the dependency to the horizon is now quadratic and not anymore exponential. Comparing this result with classic imitation learning we have worse dependency on $H$, since in the classic setting is only linear (we can recover this dependency when $L_P = 0$). However, as explained before, the fact that the transition kernel does depend on the population indeed intuitively implies a larger amplification of errors. Sec. 5 provides empirical evidence that this approach is better than the previous two, with much less influence from $H$ and $L_P$.

Now, we provide an adversarial formulation, to get a sense of what a practical algorithm could look like. Using again an IPM argument, we have that (details in Appx. B)

$$\min_\pi \sum_{n=0}^{H-1} \|\mu_n^{(\pi^E)\pi^E} - \mu_n^{(\pi)\pi}\|_1 = \min_\pi \max_{f\in\mathcal{R}} \left\{V_f(\pi^E, \rho^{(\pi^E)}) - V_f(\pi, \rho^{(\pi)})\right\}.$$

Notice that it is not obvious if we can switch the min and the max here, due to the underlying set of policy-induced occupancy measures being not necessarily convex (due to the dependency of the dynamics on the population). Assuming we can, we still learn an intermediate population-independent non-stationary reward, but now the underlying control problem is no longer RL, it is an MFC problem, as it implies studying $\max_\pi V_f(\pi, \rho^{(\pi)})$, where the population does depend on the optimized policy. This suggests that in practice one could start from a classic adversarial IL approach, and replace the underlying RL algorithm by an inner MFC algorithm (e.g. [28, 7, 21, 17, 3]). We leave the design and implementation of such a practical algorithm for future work, which may be more complex than a straight replacement of the control part (notably, many MFC approaches learn a population-dependent policy). However, this overall suggests that making progress on IL for MFGs may have to build upon MFC.

## 5 Simulation

In order to provide some empirical evidences of the insights given by our analysis (influence of the horizon and the dependency of the dynamics to the population on the various considered proxys to the Nash imitation gap), we introduce the "Attractor MFG" (see more details about it in App. D). This is a 2-state and 2-action MFG with initial distribution satisfying $\rho_0(s_0) = 1$, with horizon $H$ and with Lipschitz parameter $L$. The reward only depends on the state (not on the distribution nor the

action) and satisfies for all $a \in \mathcal{A}$, $\rho \in \Delta_{\mathcal{S}}$,

$$r(s_0, a, \rho) = 0 \text{ and } r(s_1, a, \rho) = -1.$$

In the state $s_1$, any choice of actions leads deterministically to $s_1$, the transition kernel satisfies $P(s_1|s_1, a, \rho) = 1$ for all $a \in \mathcal{A}$, $\rho \in \Delta_{\mathcal{S}}$. In the state $s_0$, the action $a_0$ leads deterministically to $s_1$, while action $a_0$ leads stochastically to one of the two states: the higher the fraction of the population in $s_1$, the higher the chance to transit to $s_1$ after choosing $a_0$:

$$P(s_1|s_0, a_1, \rho) = 1 \text{ and } P(s_1|s_0, a_0, \rho) = \min\{1, L\rho(s_1)\}.$$

Therefore, the state $s_1$ is an attractor, hence the chosen name for the MFG.

The experiment consists in computing the errors $\epsilon_n^{\mathrm{BC}}$, $\epsilon_n^{\mathrm{vanilla-ADV}}$, and $\epsilon_n^{\mathrm{MFC-ADV}}$ for various values of $L = \{0.01, 0.05, 0.1, 0.5\}$ and $H = \{3, 25, 50, 75, 100\}$ and we show the NIG as a function of the mentioned errors. From Fig. 1, we can observe an exponential dependency of the horizon for both BC and vanilla-ADV when $L$ increases, while this does not happen for MFC-ADV. This experiment demonstrates empirically what our theoretical upper bound shows, supporting the insights of our analysis. This suggests that a practical algorithm for IL in MFGs needs to minimize the MFC-ADV error (see Section 4.2). In Appx. D we provide more experimental results.

## 6   Conclusion

In this paper, we have studied the recent question of imitation learning in mean-field games. We have reviewed the few previous works tackling this problem and provided a critical discussion on the proposed approach. We then have introduced the new solution concept of Nash imitation gap to quantify the quality of imitation. In the simpler case of a population-independent dynamics, we have shown that the problem basically reduces to single-agent imitation learning, and that abstractions of the canonical BC and adversarial approaches come with a similar performance guarantee. In the harder population-dependent case, we have provided upper-bounds that are *exponential* in the horizon, for both BC and adversarial IL, the latter being the approach adopted in previous work. We also introduce a new proxy that amounts to control different occupancy measures, not solely driven by the expert population but by the policy-induced population. From a practical viewpoint, it only implies interacting with the MFG (without access to the expert population for driving the dynamics), and from a theoretical viewpoint it enjoys a much better quadratic dependency to the horizon. The associated adversarial formulation suggests that classic adversarial IL approaches could be adapted by replacing the inner RL loop by an MFC one.

This work opens a number of interesting research questions. Our last result suggests a simple modification of existing adversarial approaches, but in practice it may be more difficult than just replacing the control part, and could call for additional research in MFC and MFGs. Maybe also that controlling other kinds of occupancy measures, or even different quantities related to the policy and dynamics, may lead to even better guarantees, ideally recovering the linear dependency to the horizon of the single-agent case. We provided upper-bounds, but there is no known associated lower-bound in this setting, which is another interesting research direction. We focused on experts achieving a Nash equilibrium, and extending our work to different kind of equilibria is another possible direction. We focused on imitating the expert, not on recovering the underlying optimized reward (inverse RL). Doing so may require a different approach, as our adversarial formulations all rely on population-independent rewards (which is convenient from a practical viewpoint, whenever one just wants to recover the policy). We also plan to work on the extension of Nash imitation gap to other kinds of games.

## Funding and Acknowledgements

Giorgia Ramponi was supported by the ETH AI Center and Google Deepmind. Pavel Kolev was supported by the Cyber Valley Research Fund and the Volkswagen Stiftung (No 98 571). Mathieu Lauriere is with the NYU Shanghai Frontiers Science Center of Artificial Intelligence and Deep Learning, and the NYU-ECNU Institute of Mathematical Sciences at NYU Shanghai. Part of the work was done while Mathieu Lauriere was a visiting faculty at Google Brain. We thank anonymous reviewers for comments, which helped improve the presentation of the paper.

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
