# Appendix

## Table of Contents

## A  Application of Pinsker's Inequality

In the MFG-IL setting with BC, the agent seeks to learn a policy $\pi^A$ that minimizes the expected KL divergence objective $\epsilon^{\mathrm{BC}} := \max_{n \in \{0,\dots,H-1\}} \epsilon_n^{\mathrm{BC}}$, where $\epsilon_n^{\mathrm{BC}} := \mathbb{E}_{s \sim \rho_n^E}[\mathrm{KL}(\pi_n^E(\cdot|s) || \pi_n^A(\cdot|s))]$. Although the agent may assign a zero probability to an action that the expert assigns a strictly positive probability, in this case $\epsilon^{\mathrm{BC}}$ equals infinity. However, this scenario is quite unlikely (e.g., if the agent policy is a softmax, a classic approach). Furthermore, using Pinsker's inequality, it can be shown that

$$\mathbb{E}_{s \sim \rho_n^E}[\|\pi_n^E(\cdot|s) - \pi_n^E(\cdot|s)\|_1] \leq \sqrt{2\mathbb{E}_{s \sim \rho_n^E}[\mathrm{KL}(\pi_n^E(\cdot|s)||\pi_n^A(\cdot|s))]} \leq \sqrt{2\epsilon^{\mathrm{BC}}}.$$

It is crucial to emphasize that substituting the $\ell_1$ BC bound in (1) with the KL divergence bound above, yields the desired BC error with $\sqrt{\epsilon}$ dependence. More precisely, apply the above inequality to (8) to update Lemma 3 and to (9) for Lemma 4. As an immediate corollary (showing an exponential gap between single agent and mean field games in imitation learning settings), the Nash imitation gap satisfies $\mathcal{E}(\pi^A) \leq H^2(2L_r + r_{\max})\sqrt{2\epsilon^{\mathrm{BC}}}$ when $L_P = 0$ (corollary of Theorem 1), otherwise $\mathcal{E}(\pi^A) \leq \left(\frac{2(L_r + r_{\max})}{L_P^2}(1 + L_P)^H + r_{\max}H^2\right)\sqrt{2\epsilon^{\mathrm{BC}}}$ when $L_P > 0$ (corollary of Theorem 3). We can easily derive similar results for the adversarial setting by considering the Jensen–Shannon divergence optimized by GAIL. Moreover, similar to the infinite-horizon single-agent IRL setting, studied by Xu et al. [32], our analysis does not require any additional assumptions.

## B  More details on IPMs

In this section, we provide a detailed derivation and additional discussions of the adversarial viewpoint of the minimization of the $\ell_1$-distance between occupancy measures. Generally speaking, for a finite set $X$ and $\Delta_X$ the associated simplex, we can express the $\ell_1$-norm between probability distributions (their total variation) as an integral probability metric (IPM). For any $p, q \in \Delta_X$, we have

$$\|p - q\|_1 = \sup_{f: X \to [-1,1]} (\mathbb{E}_{x \sim p}[f(x)] - \mathbb{E}_{x \sim q}[f(x)]).$$

This will be the building block for framing an adversarial formulation of imitation learning, and making links to classic approaches such as GAIL, even if they consider usually a different framework (single agent with $\gamma$-discounted infinite horizon).

## B.1 Classic imitation setting

First, we consider the classic IL setting depicted in Sec. 2.2. Let $f : \mathcal{S} \times \mathcal{A} \times [H-1] \to \mathbb{R}$ be a (non-stationary) reward, we recall that the value function can be written as

$$V_f(\pi) = \sum_{s,a} \mu_n^\pi(s,a) f_n^\pi(s,a).$$

We claimed that adversarial IL can be framed as minimizing for all time step $n$ the distance $\|\mu_n^E - \mu_n^A\|$. To see this, for a policy $\pi$ with associated sequence of occupancy measures $\mu_0^\pi, \dots, \mu_{H-1}^\pi$, define

$$\tilde{\mu}^\pi = \frac{1}{H} \begin{pmatrix} \mu_0^\pi & \cdots & \mu_{H-1}^\pi \end{pmatrix} \in \Delta_{\mathcal{S} \times \mathcal{A} \times [H-1]}.$$

Recall the set $\mathcal{R}$ defined in Sec. 2.1, $\mathcal{R} = \{\mathcal{S} \times \mathcal{A} \times [H-1] \to [-1,1]\}$. Using the IPM viewpoint, we can write

$$\|\tilde{\mu}^E - \tilde{\mu}^\pi\|_1 = \max_{f \in \mathcal{R}} (\mathbb{E}_{s,a \sim \tilde{\mu}_n^E}[f_n(s,a)] - \mathbb{E}_{s,a \sim \tilde{\mu}_n^\pi}[f_n(s,a)])$$

$$= \max_{f \in \mathcal{R}} \frac{1}{H} (\sum_{s,a} \mu_n^E(s,a) f_n(s,a) - \sum_{s,a} \mu_n^\pi(s,a) f_n(s,a))$$

$$= \max_{f \in \mathcal{R}} \frac{1}{H} (V_f(\pi^E) - V_f(\pi)).$$

Therefore, we can frame the imitation learning problem as finding a non-stationary policy $\pi^A \in \arg\min_\pi \|\tilde{\mu}^E - \tilde{\mu}^\pi\|_1$, which amounts to solve

$$\min_\pi \sum_{n=0}^{H-1} \|\mu_n^E - \mu_n^\pi\|_1 = \min_\pi H \|\tilde{\mu}^E - \tilde{\mu}^\pi\|_1$$

$$= \min_\pi \max_{f \in \mathcal{R}} (V_f(\pi^E) - V_f(\pi))$$

$$= \max_{f \in \mathcal{R}} \min_\pi (V_f(\pi^E) - V_f(\pi)),$$

where the last equation holds because the saddle-point objective is linear in both $f$ and $\tilde{\mu}^\pi$. This is the result claimed in Sec. 2.2.

This is reminiscent of the classic adversarial approaches of the literature, with the inner problem consisting in solving a (non-stationary here) RL problem. There are important differences: the usual framework is $\gamma$-discounted infinite horizon, and as far as we know no practical approach is based on the IPM of the total variation. Rather, many of these adversarial approaches can be framed as minimizing an $f$-divergence between occupancy measures [15, 19], for example GAIL minimize an entropy-regularized Jensen-Shannon divergence [18]. We think that considering the total variation for our analysis and exposition is a practical and meaningful abstraction: it allows providing an analysis, and it could be an inspiration for deriving practical algorithms by applying a similar recipe.

## B.2 MFG adversarial imitation when $L_P = 0$

When $L_P = 0$, the transition kernel of the MFG does not depend on the population. The reward does, but from the imitating agent viewpoint, there is an expert policy to imitate and an MDP without reward (the MFG without reward) to interact with. In other words, the minimization of the distance can be framed exactly as in the previous section:

$$\min_\pi \sum_{n=0}^{H-1} \|\mu_n^E - \mu_n^\pi\|_1 = \max_{f \in \mathcal{R}} \min_\pi (V_f(\pi^E) - V_f(\pi)).$$

So, no matter whether the expert policy is at a Nash equilibrium in an MFG or not, from the imitating agent this can be framed as a reduction to classical single agent imitation learning, with the exactly same guarantee.

## B.3 MFG adversarial imitation when $L_P > 0$

First, consider the adversarial imitation approach studied in Sec. 4.2, that is we assume to control $\|\mu_n^{(\pi^E)\pi^E} - \mu_n^{(\pi^E)\pi^A}\|_1$. In essence, this means that we fix the population to be the expert one and ask a representative agent to imitate the expert policy. However, if we fix the population, the MFG without reward reduces to an MDP without reward, and we are again in the same case. We have

$$\|\tilde{\mu}^{(\pi^E)\pi^E} - \tilde{\mu}^{(\pi^E)\pi}\|_1 = \max_{f \in \mathcal{R}} (\mathbb{E}_{s,a \sim \tilde{\mu}_n^{(\pi^E)\pi^E}}[f_n(s,a)] - \mathbb{E}_{s,a \sim \tilde{\mu}_n^{(\pi^E)\pi}}[f_n(s,a)])$$

$$= \max_{f \in \mathcal{R}} \frac{1}{H}(\sum_{s,a} \mu_n^{(\pi^E)\pi^E}(s,a)f_n(s,a) - \sum_{s,a} \mu_n^{(\pi^E)\pi}(s,a)f_n(s,a))$$

$$= \max_{f \in \mathcal{R}} \frac{1}{H}(V_f(\pi^E, \rho^{(\pi^E)}) - V_f(\pi, \rho^{(\pi^E)})).$$

From this, as before we can deduce that

$$\min_{\pi} \sum_{n=0}^{H-1} \|\mu_n^{(\pi^E)\pi^E} - \mu_n^{(\pi^E)\pi}\|_1 = \max_{f \in \mathcal{R}} \min_{\pi}(V_f(\pi^E, \rho^{(\pi^E)}) - V_f(\pi, \rho^{(\pi^E)})).$$

The population being fixed to the expert one in both value functions, this is again a reduction to classic imitation, and the dual variable is a non-stationary reward that does not need to depend on the population. In other words, the inner problem is again a (non-stationary) RL problem. However, as discussed in Sec. 4.2, this does not come with encouraging theoretical guarantees, due to the exponential dependency on the horizon.

Eventually, let us consider the case of Sec. 4.3, that is, we assume to control $\|\mu_n^{(\pi^E)\pi^E} - \mu_n^{(\pi^A)\pi^A}\|_1$. Here, we no longer have a reduction to classic adversarial IL, because the two occupancy measures depends on different populations, but we can still obtain an adversarial formulation using the same IPM viewpoint. We have

$$\|\tilde{\mu}^{(\pi^E)\pi^E} - \tilde{\mu}^{(\pi)\pi}\|_1 = \max_{f \in \mathcal{R}} (\mathbb{E}_{s,a \sim \tilde{\mu}_n^{(\pi^E)\pi^E}}[f_n(s,a)] - \mathbb{E}_{s,a \sim \tilde{\mu}_n^{(\pi)\pi}}[f_n(s,a)])$$

$$= \max_{f \in \mathcal{R}} \frac{1}{H}(\sum_{s,a} \mu_n^{(\pi^E)\pi^E}(s,a)f_n(s,a) - \sum_{s,a} \mu_n^{(\pi)\pi}(s,a)f_n(s,a))$$

$$= \max_{f \in \mathcal{R}} \frac{1}{H}(V_f(\pi^E, \rho^{(\pi^E)}) - V_f(\pi, \rho^{(\pi)})).$$

From this, as before we can deduce that

$$\min_{\pi} \sum_{n=0}^{H-1} \|\mu_n^{(\pi^E)\pi^E} - \mu_n^{(\pi)\pi}\|_1 = \min_{\pi} \max_{f \in \mathcal{R}}(V_f(\pi^E, \rho^{(\pi^E)}) - V_f(\pi, \rho^{(\pi)})).$$

Notice that here, it is not obvious to know if we can switch the min and the max. Indeed, for this to hold, we need the set of policy-induced occupancy measures to be a convex set (in addition to the linearity of the value in both the reward and the occupancy measure). Whenever the dynamics does not depend on the population, this is true, this set is even a polytope. When the dynamics depends on the population, it is less clear, and ensuring the convexity of the underlying set may require additional assumptions on the transition kernel. We leave this interesting question for future work. For now, we assume that we can switch the min and the max, even if heuristically.

In this case, the underlying control problem is no longer an RL problem, but an MFC problem, as it implies solving for $\max_\pi V_f(\pi, \rho^{(\pi)})$, with again the reward being non-stationary, but still population-independent. This suggests that for obtaining such an adversarial IL approach for MFGs, one could start from an existing adversarial approach for the classic setting (for example, GAIL), and replace the underlying RL optimization problem by an MFC optimization problem. We leave the development of a more practical algorithm for future work, and it would probably call for more than just plugging an MFC algorithm in GAIL, but this suggests that making progress in IL for MFGs may have to build upon MFC.

# C Proofs of stated theoretical results

In this section we report the proof of the theorems written in the paper. The main idea of the proof is to decompose the exploitability error.

## C.1 Decomposition of the exploitability

In this subsection we provide the decomposition of the Nash imitation gap, that is the exploitability. For now, we do not make any assumption on how the policy $\pi^A$ is obtained; it can be any policy. Our goal is to decompose:

$$\mathcal{E}(\pi^A) = \max_{\pi'} V(\pi', \rho^A) - V(\pi^A, \rho^A),$$

where we recall that we write $\rho^A$ as a shorthand for $\rho^{(\pi^A)\pi^A}$, and similarly for other quantities (see Sec. 2.2). Then, proving our main results will amount to bound each term of the decomposition, depending on the setting ($L_P = 0$ or $L_P > 0$) and the kind of considered error.

Instead of bounding the exploitability, we bound the value difference for any policy $\pi'$ (the bound on the exploitability is a simple corollary by maximizing over $\pi'$).

**Lemma 1.** *Under Asm. 1 and 2, for any policies $\pi^A$ and $\pi'$, we have*

$$|V(\pi', \rho^A) - V(\pi^A, \rho^A)| \le 2L_r \sum_{n=0}^{H-1} \|\rho_n^A - \rho_n^E\|_1$$

$$+ r_{max} \left( \sum_{n=0}^{H-1} \|\mu_n^{(\pi^E)\pi^E} - \mu_n^{(\pi^E)\pi^A}\|_1 + \sum_{n=0}^{H-1} \|\rho_n^{(\pi^A)\pi'} - \rho_n^{(\pi^E)\pi'}\|_1 + \sum_{n=0}^{H-1} \|\rho_n^{(\pi^A)\pi^A} - \rho_n^{(\pi^E)\pi^A}\|_1 \right).$$

*Proof.* We start by decomposing the value difference:

$$V(\pi', \rho^A) - V(\pi^A, \rho^A) = \underbrace{V(\pi', \rho^A) - V(\pi^E, \rho^E)}_{A} + \underbrace{V(\pi^E, \rho^E) - V(\pi^A, \rho^A)}_{B}.$$

The idea is to study the distance between the value function with the quantity of interest, that is the Nash equilibrium policy $\pi^E$.

**Term A.** We decompose the term $A$ again, summing and subtracting $V(\pi', \rho^E)$:

$$A = V(\pi', \rho^A) - V(\pi^E, \rho^E)$$
$$= \underbrace{V(\pi', \rho^A) - V(\pi', \rho^E)}_{A_1} + \underbrace{V(\pi', \rho^E) - V(\pi^E, \rho^E)}_{A_2}.$$

The term $A_2$ can be interpreted as the gain we have using a different policy fixing the Nash equilibrium distribution $\rho^E$. Since $(\pi^E, \rho^E)$ is a Nash equilibrium by assumption, then $\mathcal{E}(\pi^E) = 0$ and so $A_2 \le 0$. Then, we need to study only the term $A_1$. Using Lemma 2 (see Appx. C.4) we have:

$$|V(\pi', \rho^A) - V(\pi', \rho^E)| \le L_r \sum_{n=0}^{H-1} \|\rho_n^A - \rho_n^E\|_1 + r_{max} \sum_{n=0}^{H-1} \|\rho_n^{(\pi^A)\pi'} - \rho_n^{(\pi^E)\pi'}\|_1$$

Therefore, we have a bound for the term $A$:

$$|A| \le L_r \sum_{n=0}^{H-1} \|\rho_n^A - \rho_n^E\|_1 + r_{max} \sum_{n=0}^{H-1} \|\rho_n^{(\pi^A)\pi'} - \rho_n^{(\pi^E)\pi'}\|_1.$$

**Term B.** We decompose the term B in the following way:

$$B = V(\pi^E, \rho^E) - V(\pi^A, \rho^A)$$
$$\le \underbrace{V(\pi^E, \rho^E) - V(\pi^A, \rho^E)}_{B_1} + \underbrace{V(\pi^A, \rho^E) - V(\pi^A, \rho^A)}_{B_2}.$$

We start by bounding the term $B_1$ (the positiveness of $B_1$ comes from $\pi_E$ being a Nash equilibrium):

$$0 \leq B_1 = V(\pi^E, \rho^E) - V(\pi^A, \rho^E)$$

$$= \sum_{n=0}^{H-1} \sum_{s,a} \left( \mu_n^{(\pi^E)\pi^E}(s,a) r(s,a,\rho^E) - \mu_n^{(\pi^A)\pi^E}(s,a) r(s,a,\rho^E) \right)$$

$$= \sum_{n=0}^{H-1} \sum_{s,a} \left( \left( \mu_n^{(\pi^E)\pi^E}(s,a) - \mu_n^{(\pi^A)\pi^E}(s,a) \right) r(s,a,\rho^E) \right)$$

$$\leq r_{\max} \sum_{n=0}^{H-1} \sum_{s,a} \left| \mu_n^{(\pi^E)\pi^E}(s,a) - \mu_n^{(\pi^A)\pi^E}(s,a) \right| \text{ (using Asm. 2)}$$

$$= r_{\max} \sum_{n=0}^{H-1} \| \mu_n^{(\pi^E)\pi^E} - \mu_n^{(\pi^A)\pi^E} \|_1.$$

For the term $B_2$ we can use again Lemma 2:

$$|V(\pi^A, \rho^E) - V(\pi^A, \rho^A)| \leq L_r \sum_{n=0}^{H-1} \| \rho_n^E - \rho_n^A \|_1 + r_{\max} \sum_{n=0}^{H-1} \| \rho^{(\pi^E)\pi^A} - \rho^{(\pi^A)\pi^A} \|_1.$$

Then, putting things together:

$$|B| \leq r_{\max} \sum_{n=0}^{H-1} \| \mu_n^{(\pi^E)\pi^E} - \mu_n^{(\pi^A)\pi^E} \|_1 + L_r \sum_{n=0}^{H-1} \| \rho_n^E - \rho_n^A \|_1 + r_{\max} \sum_{n=0}^{H-1} \| \rho^{(\pi^E)\pi^A} - \rho^{(\pi^A)\pi^A} \|_1.$$

**Final bound.** Applying the triangle inequality on the absolute value of the initial decomposition and injecting the bounds of $|A|$ and $|B|$, we obtain the stated result:

$$|V(\pi', \rho^A) - V(\pi^A, \rho^A)| \leq 2L_r \sum_{n=0}^{H-1} \| \rho_n^A - \rho_n^E \|_1$$

$$+ r_{\max} \left( \sum_{n=0}^{H-1} \| \mu_n^{(\pi^E)\pi^E} - \mu_n^{(\pi^E)\pi^A} \|_1 + \sum_{n=0}^{H-1} \| \rho_n^{(\pi^A)\pi'} - \rho_n^{(\pi^E)\pi'} \|_1 + \sum_{n=0}^{H-1} \| \rho_n^{(\pi^A)\pi^A} - \rho_n^{(\pi^E)\pi^A} \|_1 \right).$$

$\square$

Notice that whenever $L_P = 0$, two terms of the above bound cancel out, $\sum_{n=0}^{H-1} \| \rho_n^{(\pi^A)\pi'} - \rho_n^{(\pi^E)\pi'} \|_1 = 0$ and $\sum_{n=0}^{H-1} \| \rho_n^{(\pi^A)\pi^A} - \rho_n^{(\pi^E)\pi^A} \|_1 = 0$, as the occupancy measure does not depend on the population. This will be useful in the next section.

### C.2 Proofs for the case $L_P = 0$

In this section we analyze the case in which the transition model is independent from the population distribution. As explained above, with the terms canceling out, the value difference is bounded as:

$$|V(\pi', \rho^A) - V(\pi^A, \rho^A)| \leq 2L_r \underbrace{\sum_{n=0}^{H-1} \| \rho_n^A - \rho_n^E \|_1}_{T_1} + r_{\max} \underbrace{\left( \sum_{n=0}^{H-1} \| \mu_n^{(\pi^E)\pi^E} - \mu_n^{(\pi^E)\pi^A} \|_1 \right)}_{T_2}.$$

We analyze now the two errors considered: the one from Behavioral Cloning and the adversarial one.

**Behavioral Cloning.** Recall the definition of the term $\epsilon_n^{\text{BC}}$ and the stated result.

$$\epsilon_n^{\text{BC}} := \mathbb{E}_{s \sim \rho_n^E} \left[ \| \pi_n^A(s) - \pi_n^E(s) \|_1 \right].$$

**Theorem 1.** *Let* $\epsilon^{\mathrm{BC}} = \max_{n \in \{0,\ldots,H-1\}} \epsilon_n^{\mathrm{BC}}$. *If* $L_P = 0$, *the Nash imitation gap satisfies*

$$\mathcal{E}(\pi^A) \le H^2 (2L_r + r_{max}) \epsilon^{\mathrm{BC}}.$$

*Proof.* Both terms $T_1$ and $T_2$ can be bounded using Lemma 3 in Appx. C.4 which connects the $\ell_1$ distance between the two policies with the state-action distribution sequence, giving the stated result as a corollary. For the term $T_1$, to apply the lemma, it may be worth emphasizing that when $L_P = 0$, we have that $\|\rho_n^A - \rho_n^E\|_1 = \|\rho_n^{(\pi^A)\pi^A} - \rho_n^{(\pi^E)\pi^E}\|_1 = \|\rho_n^{(\pi^E)\pi^A} - \rho_n^{(\pi^E)\pi^E}\|_1$. $\square$

**Adversarial learning.** Recall the definition of the term $\epsilon_n^{\mathrm{ADV}}$ and the stated result.

$$\epsilon_n^{\mathrm{ADV}} := \|\mu_n^{\pi^A} - \mu_n^{\pi^E}\|_1, \qquad n \in \{0,\ldots,H-1\}.$$

**Theorem 2.** *Let* $\epsilon^{\mathrm{ADV}} = \max_{n \in \{0,\ldots,H-1\}} \epsilon_n^{\mathrm{ADV}}$. *If* $L_P = 0$, *the Nash imitation gap satisfies*

$$\mathcal{E}(\pi^A) \le H(2L_r + r_{\max}) \epsilon^{\mathrm{ADV}}.$$

*Proof.* We start by bounding the term $T_1$.

$$
\begin{aligned}
\|\rho_n^A - \rho_n^E\|_1 &= \sum_s |\rho_n^A(s) - \rho_n^E(s)| \\
&= \sum_s \left| \sum_a \mu_n^A(s,a) - \mu_n^E(s,a) \right| \\
&\le \sum_{s,a} |\mu_n^A(s,a) - \mu_n^E(s,a)| = \|\mu_n^E - \mu_n^A\|_1 \le \epsilon^{\mathrm{ADV}},
\end{aligned}
$$

and thus:

$$\sum_{n=0}^{H-1} \|\rho_n^A - \rho_n^E\|_1 \le H\epsilon^{\mathrm{ADV}}.$$

The second term, $T_2$, is bounded by $H\epsilon^{\mathrm{ADV}}$ by definition of $\epsilon^{\mathrm{ADV}}$. Then putting things together we recover the stated result. $\square$

### C.3 Proofs for the case $L_P > 0$

We report in this section the results for the more general case in which the transition dynamics depends on the population. We recall the bound on the value difference given by Lemma 1:

$$
|V(\pi', \rho^E) - V(\pi^E, \rho^E)| \le 2L_r \underbrace{\sum_{n=0}^{H-1} \|\rho_n^A - \rho_n^E\|_1}_{T_1} + r_{\max} \left( \underbrace{\sum_{n=0}^{H-1} \|\mu_n^{(\pi^E)\pi^E} - \mu_n^{(\pi^E)\pi^A}\|_1}_{T_2} \right.
$$
$$
\left. + \underbrace{\sum_{n=0}^{H-1} \|\rho_n^{(\pi^A)\pi'} - \rho_n^{(\pi^E)\pi'}\|_1}_{T_3} + \underbrace{\sum_{n=0}^{H-1} \|\rho_n^{(\pi^A)\pi^A} - \rho_n^{(\pi^E)\pi^A}\|_1}_{T_4} \right).
$$

**Behavioral cloning.** Recall the definition of the term $\epsilon_n^{\mathrm{BC}}$ and the stated result.

$$\epsilon_n^{\mathrm{BC}} := \mathbb{E}_{s \sim \rho_n^E} \left[ \|\pi_n^A(s) - \pi_n^E(s)\|_1 \right].$$

**Theorem 3.** *Let* $\epsilon^{\mathrm{BC}} = \max_{n \in \{0,\ldots,H-1\}} \epsilon_n^{\mathrm{BC}}$. *If* $L_P > 0$, *the Nash imitation gap satisfies:*

$$\mathcal{E}(\pi^A) \le \left( \frac{2(L_r + r_{\max})}{L_P^2} (1 + L_P)^H + r_{max} H^2 \right) \epsilon^{\mathrm{BC}}.$$

*Proof.* We start by bounding the term $T_1$. Now, the two sequences of involved occupancy measures differ by their underlying policy, as for the case $L_P = 0$, but also by their underlying dynamics, driven by different populations. The bound of the term $T_1$ is given by Lemma 4 in Appx. C.4:

$$T_1 = \sum_{n=0}^{H-1} \|\rho_n^A - \rho_n^E\|_1 \leq \frac{(1+L_P)^H}{L_P^2}\epsilon^{\mathrm{BC}}.$$

Next, we consider the term $T_2$. The two sequences of involved occupancy measures differ by their underlying policies, but they share the same dynamics, driven by the expert population. Therefore, as for the case $L_P = 0$, we can apply Lemma 3 (see Appx. C.4), and obtain

$$T_2 = \sum_{n=0}^{H-1} \|\mu_n^{(\pi^E)\pi^E} - \mu_n^{(\pi^E)\pi^A}\|_1 \leq H^2\epsilon^{\mathrm{BC}}.$$

Eventually, we consider the terms $T_3$ and $T_4$. They have in common that the two sequences of involved occupancy measures differ by their underlying dynamics (driven by different populations), but have the same underlying policy. Both terms can be bounded as a direct corollary of Lemma 5 in Appx. C.4, by instantiating this common policy. The resulting bounds are:

$$T_3 = \sum_{n=0}^{H-1} \|\rho_n^{(\pi^A)\pi'} - \rho_n^{(\pi^E)\pi'}\|_1 \leq \frac{(1+L_P)^H}{L_P^2}\epsilon^{\mathrm{BC}},$$

$$T_4 = \sum_{n=0}^{H-1} \|\rho_n^{(\pi^A)\pi^A} - \rho_n^{(\pi^E)\pi^A}\|_1 \leq \frac{(1+L_P)^H}{L_P^2}\epsilon^{\mathrm{BC}}.$$

Putting everything together we obtain the stated bound (noticing that the bound does not depend on the policy $\pi'$, so a bound on the value difference readily gives a bound on the exploitability of $\pi^A$, that is the Nash imitation gap). $\square$

**Vanilla-ADV.** Recall the definition of the term $\epsilon_n^{\text{vanilla-ADV}}$ and the stated result.

$$\epsilon_n^{\text{vanilla}-\text{ADV}} := \|\mu_n^{(\pi^E)\pi^E} - \mu_n^{(\pi^E)\pi^A}\|_1 \qquad \forall n \in [0, \ldots, H-1].$$

**Theorem 4.** *Let* $\epsilon^{\text{vanilla}-\text{ADV}} = \max_{n\in\{0,\ldots,H-1\}} \epsilon_n^{\text{vanilla}-\text{ADV}}$. *If* $L_P > 0$ *the NE imitation gap satisfies:*

$$\mathcal{E}(\pi^A) \leq \left(\frac{2(L_r + r_{\max})}{L_P}(1+L_P)^H + r_{\max}H\right)\epsilon^{\text{vanilla}-\text{ADV}}.$$

*Proof.* We start by bounding the term $T_1$. We have that:

$$\|\rho_n^A - \rho_n^E\|_1 = \sum_s |\sum_a \mu_n^{(\pi^E)\pi^E}(s,a) - \mu_n^{(\pi^A)\pi^A}(s,a)|$$

$$\leq \underbrace{\|\mu_n^{(\pi^E)\pi^E} - \mu_n^{(\pi^E)\pi^A}\|_1}_{\leq\epsilon^{\text{vanilla}-\text{ADV}} \text{ by def.}} + \underbrace{\|\mu_n^{(\pi^E)\pi^A} - \mu_n^{(\pi^A)\pi^A}\|_1}_{=\|\rho_n^{(\pi^E)\pi^A} - \rho_n^{(\pi^A)\pi^A}\|_1 \text{ (same policy)}}$$

$$\leq \epsilon^{\text{vanilla}-\text{ADV}} + \|\rho_n^{(\pi^E)\pi^A} - \rho_n^{(\pi^A)\pi^A}\|. \tag{5}$$

From Eq. (11), an intermediate result of the proof of Lemma 5, we have the following inequality, for any policy $\pi$:

$$\|\rho_{n+1}^{(\pi^A)\pi} - \rho_{n+1}^{(\pi^E)\pi}\|_1 \leq L_P\|\rho_n^A - \rho_n^E\| + \|\rho_n^{(\pi^A)\pi} - \rho_n^{(\pi^E)\pi}\|_1.$$

Instantiating this inequality with $\pi = \pi^A$ and injecting Eq. (5), we obtain

$$\|\rho_{n+1}^{(\pi^A)\pi^A} - \rho_{n+1}^{(\pi^E)\pi^A}\|_1 \le L_P\|\rho_n^A - \rho_n^E\| + \|\rho_n^{(\pi^A)\pi^A} - \rho_n^{(\pi^E)\pi^A}\|_1$$

$$\le L_P(\epsilon^{\text{vanilla}-\text{ADV}} + \|\rho_n^{(\pi^E)\pi^A} - \rho_n^{(\pi^A)\pi^A}\|) + \|\rho_n^{(\pi^A)\pi^A} - \rho_n^{(\pi^E)\pi^A}\|_1$$

$$= L_P\epsilon^{\text{vanilla}-\text{ADV}} + (1 + L_P)\|\rho_n^{(\pi^A)\pi^A} - \rho_n^{(\pi^E)\pi^A}\|_1$$

$$\le L_P\epsilon^{\text{vanilla}-\text{ADV}}\sum_{k=0}^{n}(1 + L_P)^k$$

$$= L_P\epsilon^{\text{vanilla}-\text{ADV}}\frac{(1 + L_P)^{n+1} - 1}{L_P}$$

$$= \epsilon^{\text{vanilla}-\text{ADV}}((1 + L_P)^{n+1} - 1).$$

Next, we can inject back this last inequality in Eq. (5) to get a bound on $\|\rho_n^A - \rho_n^E\|_1$, and then sum to obtain the bound on $T_1$.

$$\|\rho_n^A - \rho_n^E\|_1 \le \epsilon^{\text{vanilla}-\text{ADV}}(1 + L_P)^n \tag{6}$$

and thus $T_1 = \sum_{n=0}^{H-1}\|\rho_n^A - \rho_n^E\|_1 \le \frac{(1 + L_P)^H - 1}{L_P}\epsilon^{\text{vanilla}-\text{ADV}} \le \frac{(1 + L_P)^H}{L_P}\epsilon^{\text{vanilla}-\text{ADV}}.$

The term $T_2$ is directly bounded by $H\epsilon^{\text{vanilla}-\text{ADV}}$, by definition of $\epsilon^{\text{vanilla}-\text{ADV}}$.

Eventually, we need to bound the terms $T_3$ and $T_4$, implying occupancy measures for different populations (thus dynamics) but the same policy. We start again from Eq. (11) from the proof of Lemma 5, for an arbitrary policy $\pi$, and inject the bound we just obtained on $\|\rho_n^A - \rho_n^E\|_1$ (see Eq. (6)):

$$\|\rho_{n+1}^{(\pi^A)\pi} - \rho_{n+1}^{(\pi^E)\pi}\|_1 \le L_P\|\rho_n^A - \rho_n^E\| + \|\rho_n^{(\pi^A)\pi} - \rho_n^{(\pi^E)\pi}\|_1$$

$$\le L_P\epsilon^{\text{vanilla}-\text{ADV}}(1 + L_P)^n + \|\rho_n^{(\pi^A)\pi} - \rho_n^{(\pi^E)\pi}\|_1$$

$$\le \epsilon^{\text{vanilla}-\text{ADV}}L_P\sum_{k=1}^{n}(1 + L_P)^k$$

$$\le \epsilon^{\text{vanilla}-\text{ADV}}(1 + L_P)^{n+1}.$$

Summing over time we obtain (as the bound does not depend on the common policy $\pi$)

$$T_3, T_4 \le \frac{(1 + L_P)^H}{L_P}\epsilon^{\text{vanilla}-\text{ADV}}.$$

Putting everything together we recover the final bound. $\qquad\square$

**MFC-ADV**  Recall the definition of the term $\epsilon_n^{\text{MFC-ADV}}$ and the stated result.

$$\epsilon_n^{\text{MFC}-\text{ADV}} = \|\mu_n^{(\pi^E)\pi^E} - \mu_n^{(\pi^A)\pi^A}\|_1 \qquad \forall n \in \{0, \dots, H-1\}.$$

**Theorem 5.** *Let* $\epsilon^{\text{MFC}-\text{ADV}} = \max_{n\in\{0,\dots,H-1\}}\epsilon_n^{\text{MFC}-\text{ADV}}$. *If* $L_P > 0$, *the Nash imitation gap satisfies:*

$$\mathcal{E}(\pi^A) \le \left[3L_Pr_{\max}H^2 + (2L_r + r_{\max})H\right]\epsilon^{\text{MFC}-\text{ADV}}.$$

*Proof.* The term $T_1$ is easily bounded by $H\epsilon^{\text{MFC}-\text{ADV}}$. Indeed, we have

$$\|\rho_n^A - \rho_n^E\|_1 = \sum_s|\rho_n^A(s) - \rho_n^E(s)|$$

$$= \sum_s|\sum_a(\mu_n^{(\pi^E)\pi^E}(s,a) - \mu_n^{(\pi^A)\pi^A}(s,a))|$$

$$\le \sum_{s,a}|\mu_n^{(\pi^E)\pi^E}(s,a) - \mu_n^{(\pi^A)\pi^A}(s,a)| = \|\mu_n^{(\pi^E)\pi^E} - \mu_n^{(\pi^A)\pi^A}\|_1$$

$$\le \epsilon^{\text{MFC}-\text{ADV}},$$

then by summing over time steps:

$$T_1 = \sum_{n=0}^{H-1} \|\rho_n^A - \rho_n^E\|_1 \le H\epsilon^{\mathrm{MFC-ADV}}.$$

We now focus on the terms $T_3$ and $T_4$. Starting again from Eq. (11), the intermediate result of the proof of Lemma 5, for an arbitrary policy $\pi$, we have:

$$\|\rho_{n+1}^{(A)\pi} - \rho_{n+1}^{(E)\pi}\|_1 \le L_P \|\rho_n^A - \rho_n^E\| + \|\rho_n^{(A)\pi} - \rho_n^{(E)\pi}\|_1.$$

Then, using the definition of $\epsilon^{\mathrm{MFC-ADV}}$ and by induction

$$\|\rho_{n+1}^{(A)\pi} - \rho_{n+1}^{(E)\pi}\|_1 \le L_P \epsilon^{\mathrm{MFC-ADV}} + \|\rho_n^{(A)\pi} - \rho_n^{(E)\pi}\|_1 \le (n+1) L_P \epsilon^{\mathrm{MFC-ADV}}. \tag{7}$$

This being true for any policy $\pi$, we obtain the bounds on $T_3$ and $T_4$ by summing:

$$T_3 = \sum_{n=0}^{H-1} \|\rho_n^{(\pi^A)\pi'} - \rho_n^{(\pi^E)\pi'}\|_1 \le H^2 L_P \epsilon^{\mathrm{MFC-ADV}},$$

$$T_4 = \sum_{n=0}^{H-1} \|\rho_n^{(\pi^A)\pi^A} - \rho_n^{(\pi^E)\pi^A}\|_1 \le H^2 L_P \epsilon^{\mathrm{MFC-ADV}}.$$

Eventually, we bound the remaining term $T_2$. We have that

$$\|\mu_n^{(\pi^E)\pi^E} - \mu_n^{(\pi^E)\pi^A}\| = \|\mu_n^{(\pi^E)\pi^E} - \mu_n^{(\pi^A)\pi^A} + \mu_n^{(\pi^A)\pi^A} - \mu_n^{(\pi^E)\pi^A}\|_1$$

$$\le \underbrace{\|\mu_n^{(\pi^E)\pi^E} - \mu_n^{(\pi^A)\pi^A}\|_1}_{\le \epsilon^{\mathrm{MFC-ADV}} \text{ by def.}} + \underbrace{\|\mu_n^{(\pi^A)\pi^A} - \mu_n^{(\pi^E)\pi^A}\|_1}_{\le n L_P \epsilon^{\mathrm{MFC-ADV}} \text{ by Eq. (7) and same policy}}$$

$$\le \epsilon^{\mathrm{MFC-ADV}} + n L_P \epsilon^{\mathrm{MFC-ADV}}.$$

Then, summing over time, we obtain

$$T_2 = \sum_{n=0}^{H-1} \|\mu_n^{(\pi^E)\pi^E} - \mu_n^{(\pi^E)\pi^A}\|_1 \le H\epsilon^{\mathrm{MFC-ADV}} + H^2 L_P \epsilon^{\mathrm{MFC-ADV}}.$$

Putting all the terms together we recover the stated bound. $\qquad\square$

### C.4 Auxiliary lemmas

In this section we report some auxiliary lemmas used in the proofs.

The first lemma bounds the value difference for a common policy but different populations.

**Lemma 2.** *For every three policies $\pi^1, \pi^2, \pi^3$ and the associated population distributions $\rho^{(\pi^1)}, \rho^{(\pi^2)}$, we have under Asm. 1 and writing here $r_{max} = \max_{s,a} |r(s, a, \rho^{(\pi^2)})|$:*

$$|V(\pi^3, \rho^{(\pi^1)}) - V(\pi^3, \rho^{(\pi^2)})| \le L_r \sum_{n=0}^{H-1} \|\rho_n^{(\pi^1)} - \rho_n^{(\pi^2)}\|_1 + r_{\max} \sum_{n=0}^{H-1} \|\rho_n^{(\pi^1)\pi^3} - \rho_n^{(\pi^2)\pi^3}\|_1.$$

**Remark 2.** *Notice that $\pi^1$ and $\pi^2$ play symmetric roles, and that we'll only call this result with $\pi^2 = \pi^E$, hence Asm. 2.*

*Proof.* In the proof, to lighten notations, we write $\rho^1 = \rho^{(\pi^1)}$ and $\rho^2 = \rho^{(\pi^2)}$. We start by decomposing the value difference as follows, by starting from the definition of the value, adding and subtracting

the term $\mu_n^{(\pi^1)\pi^3}(s,a)r(s,a,\rho_n^2)$, and using the triangle inequality:

$$|V(\pi^3,\rho^1) - V(\pi^3,\rho^2)|$$

$$= \left| \sum_{n=0}^{H-1} \sum_{s,a} \left( \mu_n^{(\pi^1)\pi^3}(s,a)r(s,a,\rho_n^1) - \mu_n^{(\pi^2)\pi^3}(s,a)r(s,a,\rho_n^2) \right) \right|$$

$$= \left| \sum_{n=0}^{H-1} \sum_{s,a} \left( \mu_n^{(\pi^1)\pi^3}(s,a)(r(s,a,\rho_n^1) - r(s,a,\rho_n^2)) + (\mu_n^{(\pi^1)\pi^3}(s,a) - \mu_n^{(\pi^2)\pi^3}(s,a))r(s,a,\rho_n^2) \right) \right|$$

$$\leq \sum_{n=0}^{H-1} \sum_{s,a} \left| \mu_n^{(\pi^1)\pi^3}(s,a)(r(s,a,\rho_n^1) - r(s,a,\rho_n^2)) \right| + \sum_{n=0}^{H-1} \sum_{s,a} \left| (\mu_n^{(\pi^1)\pi^3}(s,a) - \mu_n^{(\pi^2)\pi^3}(s,a))r(s,a,\rho_n^2) \right|.$$

We have two terms in the previous bound, and we upper-bound each of them. For the first one:

$$\sum_{n=0}^{H-1} \sum_{s,a} \left| \mu_n^{(\pi^1)\pi^3}(s,a)(r(s,a,\rho_n^1) - r(s,a,\rho_n^2)) \right| = \sum_{n=0}^{H-1} \sum_{s,a} \mu_n^{(\pi^1)\pi^3}(s,a)|(r(s,a,\rho_n^1) - r(s,a,\rho_n^2))|$$

$$\leq \sum_{n=0}^{H-1} \sum_{s,a} \mu_n^{(\pi^1)\pi^3}(s,a)L_r\|\rho_n^1 - \rho_n^2\|_1$$

$$= L_r \sum_{n=0}^{H-1} \|\rho_n^1 - \rho_n^2\|_1,$$

where the inequality is due to the Lipschitz assumption (Asm. 1).

For the second term to be bounded, we have:

$$\sum_{n=0}^{H-1} \sum_{s,a} \left| (\mu_n^{(\pi^1)\pi^3}(s,a) - \mu_n^{(\pi^2)\pi^3}(s,a))r(s,a,\rho_n^2) \right| \leq r_{\max} \sum_{n=0}^{H-1} \sum_{s,a} |\mu_n^{(\pi^1)\pi^3}(s,a) - \mu_n^{(\pi^2)\pi^3}(s,a)|$$

$$= r_{\max} \sum_{n=0}^{H-1} \sum_{s,a} \pi_n^3(a|s)|\rho_n^{(\pi^1)\pi^3}(s) - \rho_n^{(\pi^2)\pi^3}(s)|$$

$$= r_{\max} \sum_{n=0}^{H-1} \|\rho_n^{(\pi^1)\pi^3} - \rho_n^{(\pi^2)\pi^3}\|_1.$$

The first line relies on the assumption that the reward is uniformly bounded the second line is by definition of the joint occupancy measure, and the last line due to the probabilities summing to 1.

Putting things together, we obtain the stated bound. □

The next lemma provides intermediate bounds for the BC error when the dynamics is solely driven by the expert population, which also applies when $L_P = 0$ (as the dependency of the dynamics to the population disappear).

**Lemma 3.** *Recall that* $\epsilon^{\text{BC}} = \max_{n \in \{0,\ldots,H-1\}} \mathbb{E}_{s \sim \rho^E}[\|\pi_n^E(\cdot|s) - \pi_n^A(\cdot|s)\|_1]$. *We have that:*

$$\sum_{n=0}^{H-1} \|\rho_n^{(\pi^E)\pi^E} - \rho_n^{(\pi^E)\pi^A}\|_1 \leq H^2\epsilon^{\text{BC}} \qquad \text{and} \qquad \sum_{n=0}^{H-1} \|\mu_n^{(\pi^E)\pi^E} - \mu_n^{(\pi^E)\pi^A}\|_1 \leq H^2\epsilon^{\text{BC}}.$$

*Proof.* We start by working on the sequence of state occupancy measures. We proceed by induction. When $n = 0$, the two distributions are identical. For $n \geq 0$, assume that $\|\rho_n^{(\pi^E)\pi^E} - \rho_n^{(\pi^E)\pi^A}\|_1 \leq$

$n\epsilon^{\mathrm{BC}}$. Then,

$$
\|\rho_{n+1}^{(\pi^E)\pi^E} - \rho_{n+1}^{(\pi^E)\pi^A}\|_1
$$

$$
= \sum_s |\rho_{n+1}^{(\pi^E)\pi^E}(s) - \rho_{n+1}^{(\pi^E)\pi^A}(s)|
$$

$$
= \sum_s |\sum_{x,a} \rho_n^{(\pi^E)\pi^E}(x)\pi_n^E(a|x)P(s|x,a,\rho^E) - \rho_n^{(\pi^E)\pi^A}(x)\pi_n^A(a|x)P(s|x,a,\rho^E)| \text{ (by def.)}
$$

$$
\leq \sum_{x,a} |\rho_n^{(\pi^E)\pi^E}(x)\pi_n^E(a|x) - \rho_n^{(\pi^E)\pi^A}(x)\pi_n^A(a|x)| \underbrace{\sum_s P(s|x,a,\rho^E)}_{=1}
$$

$$
= \sum_{x,a} |\rho_n^{(\pi^E)\pi^E}(x)\pi_n^E(a|x) - \rho_n^{(\pi^E)\pi^E}(x)\pi_n^A(a|x) + \rho_n^{(\pi^E)\pi^E}(x)\pi_n^A(a|x) - \rho_n^{(\pi^E)\pi^A}(x)\pi_n^A(a|x)|
$$

$$
\leq \sum_x \rho_n^{(\pi^E)\pi^E}(x) \sum_a |\pi_n^E(a|x) - \pi_n^A(a|x)| + \sum_x |\rho_n^{(\pi^E)\pi^E}(x) - \rho_n^{(\pi^E)\pi^A}(x)| \underbrace{\sum_a \pi_n^A(a|x)}_{=1}
$$

$$
= \mathbb{E}_{x\sim\rho_n^E}\left[\|\pi_n^E(\cdot|x) - \pi_n^A(\cdot|x)\|_1\right] + \|\rho_n^{(\pi^E)\pi^E} - \rho_n^{(\pi^E)\pi^A}\|_1 \tag{8}
$$

$$
\leq \epsilon^{\mathrm{BC}} + \|\rho_n^{(\pi^E)\pi^E} - \rho_n^{(\pi^E)\pi^A}\|_1
$$

$$
\leq (n+1)\epsilon^{\mathrm{BC}}.
$$

$$
\|\rho_n^{(\pi^E)\pi^A}\|_1
$$

Then, summing over time provides the stated result:

$$
\sum_{n=0}^{H-1} \|\rho_n^{(\pi^E)\pi^E} - \rho_n^{(\pi^E)\pi^A}\|_1 \leq \epsilon^{\mathrm{BC}} \sum_{n=0}^{H-1} n \leq H^2 \epsilon^{\mathrm{BC}}.
$$

Building upon the previous result, we now work on the sequence of state-action occupancy measures.

$$
\|\mu_n^{(\pi^E)\pi^E} - \mu_n^{(\pi^E)\pi^A}\|_1
$$

$$
= \sum_{s,a} |\mu_n^{(\pi^E)\pi^E}(s,a) - \mu_n^{(\pi^E)\pi^A}(s,a)|
$$

$$
= \sum_{s,a} |\rho_n^{(\pi^E)\pi^E}(s)\pi^E(a|s) - \rho_n^{(\pi^E)\pi^A}(s)\pi^A(a|s)|
$$

$$
= \sum_{s,a} |\rho_n^{(\pi^E)\pi^E}(s)\pi^E(a|s) - \rho_n^{(\pi^E)\pi^E}(s)\pi^A(a|s) + \rho_n^{(\pi^E)\pi^E}(s)\pi^A(a|s) - \rho_n^{(\pi^E)\pi^A}(s)\pi^A(a|s)|
$$

$$
\leq \mathbb{E}_{s\sim\rho_n^E}[\|\pi_n^E(\cdot|s) - \pi_n^A(\cdot|s)\|_1] + \|\rho_n^{(\pi^E)\pi^E} - \rho_n^{(\pi^E)\pi^A}\|_1
$$

$$
\leq \epsilon^{\mathrm{BC}} + n\epsilon^{\mathrm{BC}} = (n+1)\epsilon^{\mathrm{BC}}.
$$

From this, by summing over time, we obtain the same stated bound. $\qquad\square$

The next technical lemma considers the propagation of errors when bounding a term involving a sequence of occupancy measures relying on both different policies and different populations (so different dynamics).

**Lemma 4.** *Recall that* $\epsilon^{\mathrm{BC}} = \max_{n\in\{0,\dots,H-1\}} \mathbb{E}_{s\sim\rho^E}[\|\pi_n^E(\cdot|s) - \pi_n^A(\cdot|s)\|_1]$ *and assume that* $L_P > 0$. *We have that:*

$$
\sum_{n=0}^{H-1} \|\rho_n^A - \rho_n^E\|_1 \leq \frac{(1+L_P)^H}{L_P^2}\epsilon^{\mathrm{BC}}.
$$

*Proof.* We will bound the term $\|\rho_{n+1}^A - \rho_{n+1}^E\|_1$. The idea is to make use of the definition of the occupancy measure to make appear both $\rho_n^A$ and $\rho_n^E$, to add and subtract various terms (namely $\rho_n^E(x)\pi_n^A(a|x)P(s|x,a,\rho_n^A)$ and $\rho_n^E(x)\pi_n^E(a|x)P(s|x,a,\rho_n^A)$), to use the triangle inequality, and then to bound each of the resulting terms.

$$\|\rho_{n+1}^A - \rho_{n+1}^E\|_1 = \sum_s |\rho_{n+1}^A(s) - \rho_{n+1}^E(s)|$$

$$= \sum_s |\sum_{x,a} \rho_n^A(x)\pi_n^A(a|x)P(s|x,a,\rho_n^A) - \rho_n^E(x)\pi_n^E(a|x)P(s|x,a,\rho_n^E)|$$

$$\leq \sum_{s,a,x} |\rho_n^A(x)\pi_n^A(a|x)P(s|x,a,\rho_n^A) - \rho_n^E(x)\pi_n^A(a|x)P(s|x,a,\rho_n^A)|$$

$$+ \sum_{s,a,x} |\rho_n^E(x)\pi_n^A(a|x)P(s|x,a,\rho_n^A) - \rho_n^E(x)\pi_n^E(a|x)P(s|x,a,\rho_n^A)|$$

$$+ \sum_{s,a,x} |\rho_n^E(x)\pi_n^E(a|x)P(s|x,a,\rho_n^A) - \rho_n^E(x)\pi_n^E(a|x)P(s|x,a,\rho_n^E)|$$

$$= \sum_x \underbrace{\sum_{s,a} \pi_n^A(a|x)P(s|x,a,\rho_n^A)|\rho_n^A(x) - \rho_n^E(x)|}_{=\|\rho_n^A - \rho_n^E\|_1}$$

$$+ \sum_{x,a} \rho_n^E(x)\underbrace{|\pi_n^E(a|x) - \pi_n^A(a|x)|}_{=\mathbb{E}_{x\sim\rho^E}[\|\pi_n^E(\cdot|x) - \pi_n^A(\cdot|x)\|_1]\leq\epsilon^{\mathrm{BC}}}\underbrace{\sum_s P(s|x,a,\rho_n^A)}_{=1} \quad (9)$$

$$+ \sum_{x,a} \rho_n^E(x)\pi_n^E(a|x)\underbrace{\sum_s |P(s|x,a,\rho_n^A) - P(s|x,a,\rho_n^E)|}_{\leq L_P\|\rho_n^A - \rho_n^E\|_1 \text{ by Asm. 1}}$$

$$\leq (1 + L_P)\|\rho_n^A - \rho_n^E\|_1 + \epsilon^{\mathrm{BC}}.$$

By direct induction, we obtain

$$\|\rho_{n+1}^A - \rho_{n+1}^E\|_1 \leq \epsilon^{\mathrm{BC}} \sum_{k=0}^{n} (1 + L_P)^k.$$

Notice that if $L_P = 0$, we retrieve the result in the proof of Lemma 3, that is $\|\rho_{n+1}^A - \rho_{n+1}^E\|_1 \leq (n+1)\epsilon^{\mathrm{BC}}$. If $L_P > 0$, this simplifies to

$$\|\rho_{n+1}^A - \rho_{n+1}^E\|_1 \leq \epsilon^{\mathrm{BC}} \frac{(1 + L_P)^{n+1} - 1}{L_P}. \quad (10)$$

Summing over time, we obtain the stated result,

$$\sum_{n=0}^{H-1} \|\rho_{n+1}^A - \rho_{n+1}^E\|_1 \leq \frac{(1 + L_P)^H}{L_P^2}\epsilon^{\mathrm{BC}}.$$

$\square$

The last technical lemma we provide considers the case when the involved sequences of occupancy measures have the same (arbitrary) policy but different populations (thus different dynamics).

**Lemma 5.** *Recall that* $\epsilon^{\mathrm{BC}} = \max_{n\in\{0,\dots,H-1\}} \mathbb{E}_{s\sim\rho^E}[\|\pi_n^E(\cdot|s) - \pi_n^A(\cdot|s)\|_1]$ *and assume that* $L_P > 0$. *Let* $\pi$ *be an arbitrary policy. We have:*

$$\sum_{n=0}^{H-1} \|\rho_n^{(A)\pi} - \rho_n^{(E)\pi}\|_1 \leq \frac{(1 + L_P)^H}{L_P^2}\epsilon^{\mathrm{BC}}.$$

*Proof.* The proof follows a similar idea as in the proof of Lemma 4. We make use of the definition of occupancy measure to make appear the measures at the previous time step, we add and subtract a

term (namely $\rho_n^{(\pi^A)\pi}(x)P(s|x,a,\rho_n^E)$ here) and use the triangle inequality, to eventually bound each of the resulting terms.

$$\|\rho_{n+1}^{(\pi^A)\pi} - \rho_{n+1}^{(\pi^E)\pi}\|_1 = \sum_s |\rho_{n+1}^{(\pi^A)\pi}(s) - \rho_{n+1}^{(\pi^E)\pi}(s)|$$

$$= \sum_s |\sum_{x,a} \rho_n^{(\pi^A)\pi}(x)\pi_n(a|x)P(s|x,a,\rho_n^A) - \rho_n^{(\pi^E)\pi}(x)\pi_n(a|x)P(s|x,a,\rho_n^E)|$$

$$\leq \sum_{s,a,x} \pi_n(a|x)|\rho_n^{(\pi^A)\pi}(x)P(s|x,a,\rho_n^A) - \rho_n^{(\pi^E)\pi}(x)P(s|x,a,\rho_n^E)|$$

$$\leq \sum_{s,a,x} \pi_n(a|x)|\rho_n^{(\pi^A)\pi}(x)P(s|x,a,\rho_n^A) - \rho_n^{(\pi^A)\pi}(x)P(s|x,a,\rho_n^E)|$$

$$+ \sum_{s,a,x} \pi_n(a|x)|\rho_n^{(\pi^A)\pi}(x)P(s|x,a,\rho_n^E) - \rho_n^{(\pi^E)\pi}(x)P(s|x,a,\rho_n^E)|$$

$$= \sum_{a,x} \rho_n^{(\pi^A)\pi}(x)\pi_n(a|x) \underbrace{\sum_s |P(s|x,a,\rho_n^A) - P(s|x,a,\rho_n^E)|}_{\leq L_P\|\rho_n^A - \rho_n^E\|_1 \text{ by Asm. 1}}$$

$$+ \underbrace{\sum_x \sum_{s,a} \pi_n(a|x)P(s|x,a,\rho_n^E)|\rho_n^{(\pi^A)\pi}(x) - \rho_n^{(\pi^E)\pi}(x)|}_{=\|\rho_n^{(\pi^A)\pi} - \rho_n^{(\pi^E)\pi}\|_1},$$

so $\|\rho_{n+1}^{(\pi^A)\pi} - \rho_{n+1}^{(\pi^E)\pi}\|_1 \leq L_P\|\rho_n^A - \rho_n^E\|_1 + \|\rho_n^{(\pi^A)\pi} - \rho_n^{(\pi^E)\pi}\|_1.$  (11)

If $L_P = 0$, this readily implies that $\|\rho_n^{(A)\pi} - \rho_n^{(E)\pi}\|_1 \leq \|\rho_0^{(A)\pi} - \rho_0^{(E)\pi}\|_1 = 0$, which is obviously the correct result (if the dynamics does not depend on the population, then the occupancy measures are the same, the underlying policy being the same).

Now consider the case of interest, $L_P > 0$. From Eq. (10) of the proof of Lemma 4, we know that

$$\|\rho_n^A - \rho_n^E\|_1 \leq \epsilon^{\text{BC}}\frac{(1+L_P)^n}{L_P}.$$

Injecting this into Eq. (11), we have

$$\|\rho_{n+1}^{(\pi^A)\pi} - \rho_{n+1}^{(\pi^E)\pi}\|_1 \leq \epsilon^{\text{BC}}(1+L_P)^n + \|\rho_n^{(\pi^A)\pi} - \rho_n^{(\pi^E)\pi}\|_1.$$

By direct induction we obtain

$$\|\rho_{n+1}^{(\pi^A)\pi} - \rho_{n+1}^{(\pi^E)\pi}\|_1 \leq \epsilon^{\text{BC}}\sum_{k=0}^n (1+L_P)^k \leq \epsilon^{\text{BC}}\frac{(1+L_P)^{n+1}}{L_P},$$

and summing over time we obtain the stated result:

$$\sum_{n=0}^{H-1} \|\rho_n^{(A)\pi} - \rho_n^{(E)\pi}\|_1 \leq \epsilon^{\text{BC}}\frac{(1+L_P)^H}{L_P^2}.$$

$\square$

## D   Numerical illustration

In order to provide some empirical evidences of the insights given by our analysis (influence of the horizon and the dependency of the dynamics to the population on the various considered proxys to the Nash imitation gap), we introduce the "Attractor MFG" depicted in Fig. 2.

This is a 2-state and 2-action MFG with initial distribution satisfying $\rho_0(s_0) = 1$, with horizon $H$ and with Lipschitz parameter $L$. The reward only depends on the state (not on the distribution nor the action) and satisfies for all $a \in \mathcal{A}, \rho \in \Delta_{\mathcal{S}}$,

$$r(s_0, a, \rho) = 0 \text{ and } r(s_1, a, \rho) = -1.$$

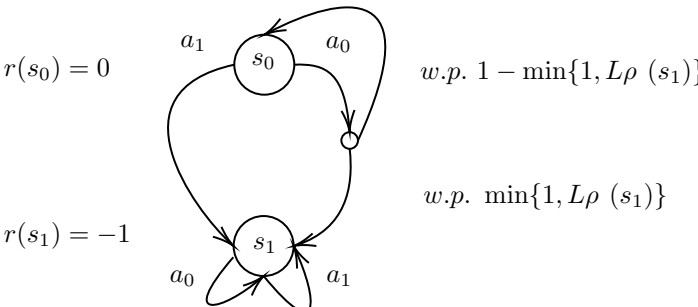

$r(s_0) = 0$    $w.p.\ 1 - \min\{1, L\rho\ (s_1)\}$

$w.p.\ \min\{1, L\rho\ (s_1)\}$

$r(s_1) = -1$

Figure 2: The "attractor" mean-field game.

In the state $s_1$, any choice of actions leads deterministically to $s_1$, the transition kernel satisfies for all $a \in \mathcal{A}, \rho \in \Delta_\mathcal{S}$,

$$P(s_1|s_1, a, \rho) = 1.$$

In the state $s_0$, the action $a_0$ leads deterministically to $s_1$, while action $a_0$ leads stochastically to one of the two states: the higher the fraction of the population in $s_1$, the higher the chance to transit to $s_1$ after choosing $a_0$:

$$P(s_1|s_0, a_1, \rho) = 1 \text{ and } P(s_1|s_0, a_0, \rho) = \min\{1, L\rho(s_1)\}.$$

Therefore, the state $s_1$ is an attractor, hence the chosen name for the MFG.

Any policy choosing action $a_0$ in state $s_0$ for every timestep is a Nash equilibrium. Denoting by $\pi^E$ such a policy, its value is $V(\pi^E, \rho^E) = 0$ (it is also a social equilibrium). The associated population obviously satisfies $\rho_n^E(s_0) = 1$ for all time steps $n = [0, \dots, H]$. We can also bound the Nash imitation gap, as any policy choosing action $a_1$ at timestep 0 in $s_0$ (and any action afterwards) will lead to the lowest possible value, that is for any policy $\pi$,

$$\mathcal{E}(\pi) \leq H - 1.$$

The Nash equilibrium being stationary, we consider the policy $\pi_\alpha$ being parameterized by the single scalar parameter $\alpha \in [0, 1]$, defined as (recall that the action selection on $s_1$ has no influence):

$$\pi^\alpha(a_1|s_0) = \alpha.$$

So, $\pi^{\alpha=0}$ is a Nash equilibrium, and $\pi^{\alpha=1}$ is a worst-case policy (of value $-(H-1)$). For such a policy, we directly get the BC error as

$$\epsilon_n^{\text{BC}}(\pi^\alpha) = \mathbb{E}_{s \sim \rho^E}[\|\pi_n^\alpha(\cdot|s) - \pi_n^E(\cdot|s)\|_1] = \|\pi_n^\alpha(\cdot|s_0) - \pi_n^E(\cdot|s_0)\|_1 = 2\alpha.$$

We can also easily compute the occupancy measures of interest by induction (it is sufficient to do so in the state $s_1$, as there are only two states):

$$\rho_0^{(E)\pi^\alpha}(s_1) = 0, \quad \rho_{n+1}^{(E)\pi^\alpha}(s_1) = \rho_n^{(E)\pi^\alpha}(s_1) + (1 - \rho_n^{(E)\pi^\alpha}(s_1))\alpha,$$

$$\rho_0^{(\pi^\alpha)}(s_1) = 0, \quad \rho_{n+1}^{(\pi^\alpha)}(s_1) = \rho_n^{(\pi^\alpha)}(s_1) + (1 - \rho_n^{(\pi^\alpha)}(s_1))(\alpha + (1 - \alpha)\min\{1, L\rho_n^{(\pi^\alpha)}(s_1)\}).$$

From this, we can easily get the related errors,

$$\epsilon_n^{\text{vanilla-ADV}}(\pi^\alpha) = \|\mu_n^{(E)} - \mu_n^{(E)\pi^\alpha}\|_1 = 2(\alpha + \rho_n^{(E)\pi^\alpha}(s_1)(1 - \alpha)),$$

$$\epsilon_n^{\text{MFC-ADV}}(\pi^\alpha) = \|\mu_n^{(E)} - \mu_n^{(\pi^\alpha)}\|_1 = 2(\alpha + \rho_n^{(\pi^\alpha)}(s_1)(1 - \alpha)).$$

With the above quantity, we also directly have the Nash imitation gap,

$$\mathcal{E}(\pi^\alpha) = \sum_{n=0}^{H-1} \rho_n^{(\pi^\alpha)}(s_1).$$

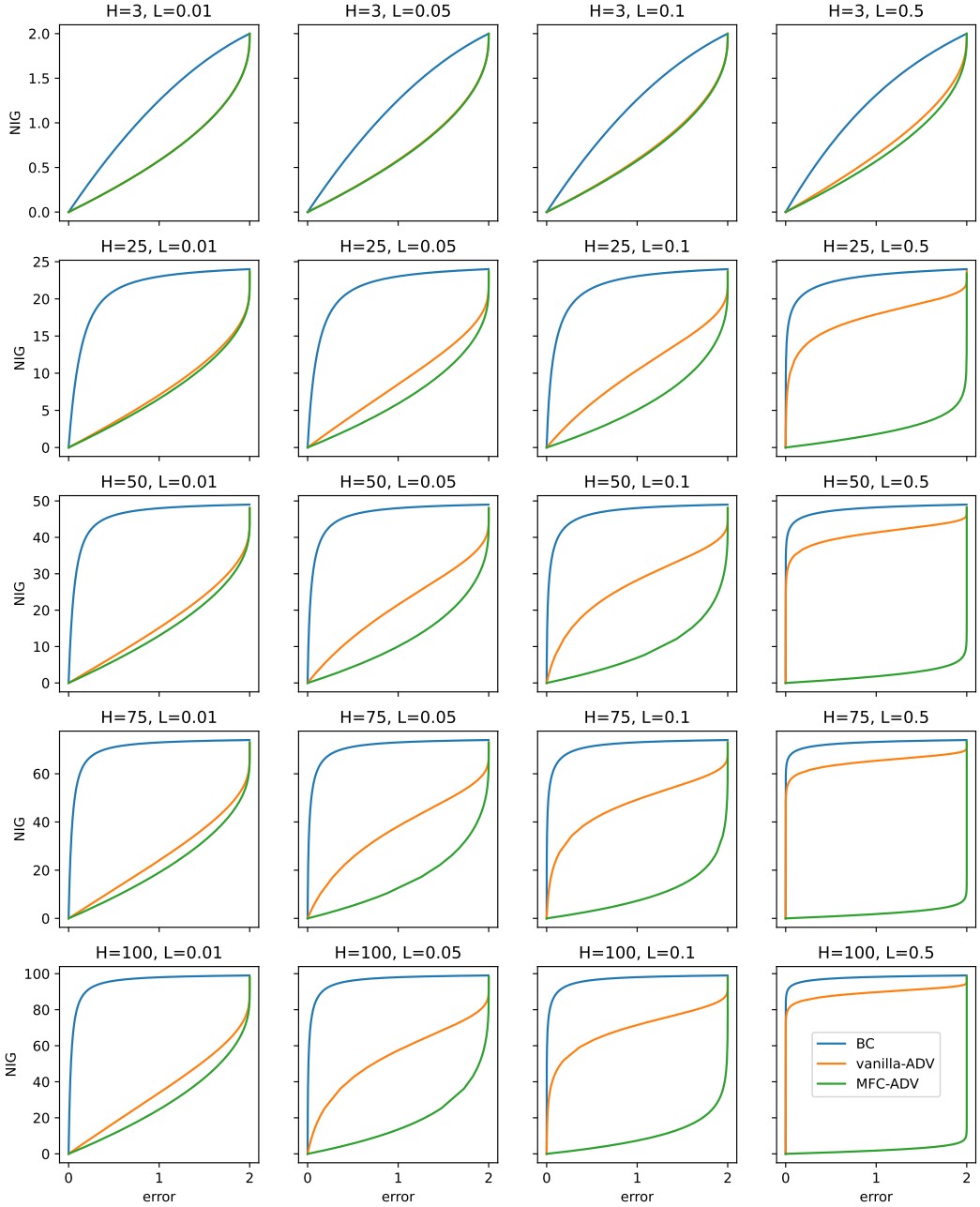

Figure 3: Nash imitation gap as a function of the considered maximum errors ($\epsilon^{\mathrm{BC}}$, $\epsilon^{\mathrm{vanilla\text{-}ADV}}$ and $\epsilon^{\mathrm{MFC\text{-}ADV}}$) in the attractor MFG, for various values of $L$ (column-wise) and $H$ (row-wise).

From this, we can also compute the maximum errors $\epsilon^{\text{BC}}(\pi_\alpha)$, $\epsilon^{\text{vanilla-ADV}}(\pi^\alpha)$ and $\epsilon^{\text{MFC-ADV}}(\pi^\alpha)$.

The numerical illustration we propose consists in computing these quantities for a grid of values of $\alpha$, for various values of $L \in \{0.01, 0.1, 0.5, 1, 2\}$ and of $H \in \{3, 25, 50, 75, 100\}$, and showing the NIG as a function of respectively $\epsilon^{\text{BC}}$, $\epsilon^{\text{vanilla-ADV}}$ and $\epsilon^{\text{MFC-ADV}}$, for the considered parameterized policies. The results are provided in Fig. 3.

We observe the following:

- NIG has a worse dependency to the BC errors than to the other ones. When either $L$ or $H$ increases, this dependency worsens, in the sense that smaller values of $\epsilon^{\text{BC}}$ are required for ensuring a given NIG.

- Vanilla-ADV and MFC-ADV behave similarly for small $L$ and $H$ (recall also that they are the same quantity for $L = 0$). However, whenever $L$ or $H$ increases, the dependency of the NIG to $\epsilon^{\text{vanilla-ADV}}$ worsens. Indeed, whenever $L$ and/or $H$ are large enough, one can observe than vanilla-ADV behaves more like BC than like MFC-ADV.

- MFC-ADV is also influenced by the values of $L$ and $H$, but much less, and is always the best approach.

Overall, this supports the insights from our analysis. In particular MFC-ADV is better than vanilla-ADV, which itself is better than BC. When $L$ and $H$ are small enough, vanilla-ADV and MFC-ADV behave similarly, and when $L$ and/or $H$ are large enough, vanilla-ADV behaves more closely to BC. Overall, this suggests that a practical IL approach for MFGs should take into account the fact that the dynamics does depend on the population, and this may build upon MFC as suggested in the main paper.