# OpenReview forum: "On Imitation in Mean-field Games"
_NeurIPS.cc/2023/Conference — NeurIPS 2023 poster_

### Official Review · Reviewer_Qrta · 2023-07-05

**Soundness:** 3 good
**Presentation:** 3 good
**Contribution:** 3 good
**Rating:** 6
**Confidence:** 3

**Summary:**

The paper presents a novel solution concept Nash imitation gap (NIG) for imitation learning problems in Mean-field games. For both population-independent and population-dependent dynamics, the author provides the upper-bounds of NIG for behavioral cloning (BC), vanilla adversarial imitation learning (vanilla-ADV), and mean-field controlling adversarial imitation learning (MFC-ADV). The empirical results show that MFC-ADV is better than vanilla-ADV which is better than BC.

**Strengths:**

Originality: The paper proposes a novel solution concept, NIG, which generalized the classic imitation gap, and shows the upper-bounds of NIG for several different settings.
Quality: The theoretical analyze is solid and supports the upper bounds well.
Clarity: The paper is clearly written and well organized.
Significance: Both the theoretical upper bounds and the empirical results suggest that the adversarial imitation learning could replace the inner RL by an MFC.


**Weaknesses:**

No lower bound of NIG is provided so that the theoretical results cannot directly indicate which methods (BC, vanilla-ADV, MFC-ADV) is the best one.

**Questions:**

1. Could the upper-bounds for BC and vanilla-ADV be improved to polynomial in the horizon? Are there any evidence showing that their NIGs should be exponential in horizon?
2. Is MFC-ADV still the best one if $s_1$ is no longer a sink in the experiment? For example, $P(s_0 \vert s_1,a_0,\rho)>0$.

---

> ### Author Rebuttal · Authors · 2023-08-08
>
> We would like to thank the reviewer for the insightful comments. We would like to thank the reviewer to recognize the originality of our paper (¨a novel solution concept, NIG, which generalized the classic imitation gap, and shows the upper-bounds of NIG for several different settings¨), and the clarity of our work and the significance (¨both the theoretical upper bounds and the empirical results suggest that the adversarial imitation learning could replace the inner RL by an MFC.¨).
> We answer the raised question below.
>
> ### Questions
>
> 1. (Also answer to Weaknesses) We provide a new figure with the rebuttal (see the 1-page pdf attached to this rebuttal), related to the experiment in Appx C. This figure shows the empirical upper-bound as a function of $H$, for this hard-instance MFG, for various values of L. One can observe that as L increases, the dependency of the (empirical) bound for both BC and Vanilla-ADV is **clearly exponential**, while that of MFC-ADV is not. This supports our theoretical analysis, and provides empirical evidence that the bound cannot be improved to be polynomial in the horizon. We plan to add this figure to the main text.
>
> 2. Choosing $P(s_0|s_1, a_0, \rho) >0$ would make the problem easier (this problem was designed specifically as being a hard MFG, hence the sink, inspired by the hard MDP construct used to showcase lower-bounds for single-agent imitation learning). However, MFC-ADV would still be better than Vanilla-ADV. The reason for this is that there will still be compounding errors at the population level (in the MFG setting, we don’t only have compounding errors at the policy level, but also at the population level, when the dynamics does depend on the population). The only way to get rid of these population-level compounding errors is to have L=0 (that depends on the problem at hand).

---

> > ### Comment · Reviewer_Qrta · 2023-08-22
> >
> > Thank the authors for the response. I still keep my score.

---

### Official Review · Reviewer_uQRX · 2023-07-06

**Soundness:** 3 good
**Presentation:** 2 fair
**Contribution:** 2 fair
**Rating:** 4
**Confidence:** 3

**Summary:**

This paper focuses on the imitation learning problem in mean-field games. The authors propose a new performance metric Nash imitation gap. They show that when the transition kernels of MDP are independent of the mean fields, the imitation learning problem reduces to that in the single-agent MDP. When the transition kernels of MDP depend on the mean fields, the upper bound of the suboptimality grows exponentially in horizon H. A new adversarial algorithm is proposed to mitigate such exponential dependency.

**Strengths:**

This paper focuses on provably efficient imitation learning algorithms. The previous works do not provide any performance guarantee for imitation learning algorithms in MFG. In contrast, the authors derive the suboptimality bounds for BC and adversarial imitation learning algorithms. Particularly, when the transition kernels depend on the mean field, the authors propose a new adversarial algorithm to mitigate the exponential dependency on the horizon.

**Weaknesses:**

1. More justifications should be provided for the l_1 norm-based algorithm. The previous works for BC all focus on the log-likelihood type algorithms, and algorithms for GAIL focus on the Jensen-Shannon divergence-based algorithm. However, this paper adopts the l_1 error optimization framework, which is quite different from the empirical implementation. More discussions should be provided to justify this new framework. In addition, how to transfer the results in this paper to the analysis of the original algorithms should be discussed.

2. As stated in the paper, when the transition kernels depend on the mean field, the exponential dependence on the horizon may not be tight. The empirical verification will be very helpful to demonstrate the true dependency on the horizon.

3. For the proposed algorithm in Section 4.3, the reformulation of the algorithm below line 353 requires the value of $\rho^{(\pi)}$. However, it is not clear how to calculate this quantity from the paper. More clarification is required here.

4. For the algorithm in Section 4.3, it is not clear why we can assume that min and max are exchangeable in the reformulation. What kind of condition is required should be explained.

**Questions:**

The questions are provided in the previous section.

**Limitations:**

Limitation part is missing in the paper.

---

> ### Author Rebuttal · Authors · 2023-08-08
>
> We would like to thank the reviewer for the insightful comments and for recognizing that our work is the first one to provide theoretical guarantees, which shows how previous approaches have exponential dependency on the horizon. Thanks also for recognizing that we ¨propose a new adversarial algorithm to mitigate the exponential dependency on the horizon¨.
> We answer the raised questions below.
>
>
> ### Weaknesses
>
> 1. ¨ *More justifications should be provided for the l_1 norm-based algorithm.* ¨ The main reason to use the L1 distance is that its simplicity conveys the main message. However, the whole analysis can be easily extended to other distance metrics (as KL-divergence). In fact, using tools such as Pinsker inequality (eg, see l.149-150), we would get similar bounds with $\sqrt{\epsilon}$ rather than $\epsilon$. This would not influence the main message of the paper which is the exponential dependency to the horizon of the BC and vanilla-ADV.
>
> 2. ¨ *The empirical verification will be very helpful to demonstrate the true dependency on the horizon.* ¨ We provide a numerical simulation in Appendix C. The experiments show that on a difficult instance, whenever the Lipschitz constant L or the horizon H are large, vanilla-ADV behaves as badly as BC, and, on the other hand, MFC-ADV continues to have a better behavior. Moreover, we provide another figure with this rebuttal (see the 1-page pdf attached to this rebuttal), that showcase the empirical upper-bound for this hard instance as a function of the horizon, where both BC and vanilla-ADV are exponential for large enough L, while MFC-ADV is not, confirming our analysis.
>
> 3. ¨ *it is not clear how to calculate this quantity from the paper.* ¨ The $\rho^\pi$ quantity can be evaluated following definition 2 (see l.88).
>
> 4. ¨ *it is not clear why we can assume that min and max are exchangeable in the reformulation.* ¨ We thank the reviewer for pointing out this relevant point. As we discussed in appendix A (557-563), we cannot always switch the min and max. In fact, we need the set of policy-induced occupancy measures to be a convex set (in addition to the linearity of the value in both the reward and the occupancy measure). Whenever the dynamics does not depend on the population, this is true, this set is even a polytope. When the dynamics depends on the population, it is less clear, and ensuring the convexity of the underlying set may require additional assumptions on the transition kernel. The MFG used in the experiment, for example, satisfies this property.

---

> > ### Comment · Reviewer_uQRX · 2023-08-17
> >
> > Thank the authors for the response. I still have the following concerns.
> >
> > 1. ¨ More justifications should be provided for the l_1 norm-based algorithm. ¨
> >
> > Since the log-likelihood optimization is much more usual in realistic applications, I think more discussion is needed. The Pinsker inequality may not be the only tool we need for other distance metrics (as KL-divergence). First, if the algorithm maximizes the log-likelihood and the aim is to derive the KL-based bound, it seems that some additional assumptions are needed. For example, the probability cannot be 0. These technical things are not discussed in the paper. Second, the Pinsker's inequality usually leads to a loose analysis in statistics, which is also known as slow rate. How to derive the tight analysis for more realistic algorithms is not mentioned in the paper.
> >
> > 2. ¨ it is not clear how to calculate this quantity from the paper. ¨
> >
> > Given the definition of $\rho^{\pi}$ in definition 2, how to calculate it in the algorithm is not clear. Concretely, definition 2 involves the knowledge of the transition kernels. In the IL environment, we usually do not have any information about the parameters of transition kernels and cannot interact with the environment. The estimation of $\mu^\pi$ in this setting seems a non-trivial problem. It is unclear how to solve this problem from the current paper.
> >
> > 3. ¨ It is not clear why we can assume that min and max are exchangeable in the reformulation. ¨
> >
> > It is not clear when we can use this algorithm, i.e., when we can exchange min and max. Besides the trivial case of distribution-independent transition kernels, a sufficient condition of the environment is needed for this exchange.

---

> > > ### Author Response · Authors · 2023-08-20
> > >
> > > Thank you for your valuable feedback and suggestions!
> > > We answer the raised concerns point by point below. We will write $\mathbb{E}$ as $E$ due to markdown formatting problems.
> > >
> > > 1. In the MFG imitation learning setting with behavior cloning, the agent seeks to learn a policy $\pi^{A}$ that minimizes the following expected KL-divergence objective:  $\epsilon^{BC} :=  \max_{n\in[H-1]} \epsilon_n^{BC}$, where  $\epsilon_n^{BC}:=E_{s \sim \rho_{n}^{E}}[\text{KL}(\pi_{n}^{E}(\cdot|s)||\pi_{n}^{A}(\cdot|s))]$. Although the agent may assign a zero probability to an action that the expert assigns a strictly positive probability, in this case the $\epsilon^{BC}$ equals infinity.
> > > However, this scenario is quite unlikely (e.g., if the agent policy is a softmax, a classic approach).
> > > Furthermore, using Pinsker's inequality, it can be shown that $E_{s\sim\rho_{n}^{E}}[\pi_{n}^{E}(\cdot|s)-\pi_{n}^{A}(\cdot|s)]\leq \sqrt{2E_{s\sim\rho_{n}^{E}}[\text{KL}(\pi_{n}^{E}(\cdot|s)||\pi_{n}^{A}(\cdot|s))]}\leq\sqrt{2\epsilon}.$ It is crucial to emphasize that substituting the $\ell_1$ BC bound in Eq.3 with the KL-divergence bound above, yields the desired BC error with $\sqrt{\epsilon}$ dependence.
> > > More precisely, apply the above inequality to line 689 to update Lemma 3 and to line 701 for Lemma 4.
> > > As an immediate corollary (showing an exponential gap between single agent and mean field games in imitation learning settings), the Nash imitation gap satisfies $\mathcal{E}(\pi^{A})\leq H^{2}(r_{\max}+2L_{r})\sqrt{2\epsilon^{BC}}$ if $L_P=0$ (Corrolary of Theorem 1), else $\mathcal{E}(\pi^{A})\leq\Big(\frac{2(L_{r}+r_{\max})}{L_{P}^{2}}(1+L_{P})^{H}+H^{2}r_{\max}\Big)\sqrt{2\epsilon^{BC}}$ when $L_P>0$ (Corrolary of Theorem 3).
> > > We can easily derive similar results for the adversarial setting by considering the JS divergence optimized by GAIL.
> > > Moreover, similar to the infinite-horizon single-agent IRL setting, studied by Xu et al. 2020 [1], our analysis does not require any additional assumptions.
> > > Following the reviewer's comments, we will add this explanation more clearly in the next version of the paper.
> > >
> > > 2. We agree with the reviewer that without access to the transition kernel, there will be errors in estimating the population distribution $\mu^{\pi}$, resulting in errors in the final recovered policy.
> > > However, notice that very few practical approaches attempt to estimate occupancy measures.
> > > Rather, an adversarial approach avoids this estimation.
> > > We explain in Section 4.3 and Appendix A how adversarial approaches could be extended to our setting by replacing the inner RL loop with an inner MFC loop, thus avoiding the need to estimate this kind of distribution.
> > > Moreover, it is crucial to emphasize that our main contribution is to show a gap between single-agent IL and MFG IL, i.e., the exponential dependence on the horizon of traditional IL methods (and also of state-of-the-art algorithms for MFG IL [2,3,4]).
> > > More specifically, we provide a systematic analysis of the MFG imitation learning setting and show which type of distance needs to be optimized, i.e., the MFC-ADV, which is different from those used in the literature.
> > > In addition, we provide a new experiment that empirically shows the exponential dependence of BC and vanilla-ADV on the horizon (see the rebuttal pdf file).
> > > While designing a practical algorithm remains an important open question, it is important to emphasize that addressing it is beyond the scope of this paper.
> > > Nevertheless, we think that the difficult part is how to deal with the MFC loop, rather than estimating the population distribution, which can be avoided thanks to an adversarial approach.
> > >
> > > 3. Thank you for raising this important point.
> > > Consider the set of populations $\rho_n$ as defined in Definition 2, induced by all stochastic policies.
> > > A sufficient condition for swapping the min and max is that this set is convex, i.e., that the transition kernel is such that, for any $\rho^1_n$ and $\rho^2_n$ in this set, then $(1-\alpha) \rho^1_n + \alpha \rho^2_n$ is also in this set, for any $\alpha\in (0,1)$.
> > > For example, this is satisfied if the transition kernel is linear in the population.
> > > It is also satisfied by the MFG we consider in Appendix C (for which the transition kernel is not linear in the population).
> > > A full characterization of transition kernels that satisfy this property is beyond the scope of this paper.
> > > Notice that this is not required for the stated theorems, only for the sketched practical adversarial approach.
> > > We would also like to underline that our main contribution is to show an exponential gap between single agent and mean-field games in the imitation learning settings.

---

> > > > ### Author Response · Authors · 2023-08-20
> > > >
> > > > Furthermore, it is crucial to emphasize that there may be other algorithmic techniques for tackling the LHS of the problem formulation in line 324, and providing a practical algorithm to solve this problem is beyond the scope of this paper.
> > > >
> > > > [1] Tian Xu, Ziniu Li, and Yang Yu. Error bounds of imitating policies and environments. Advances in Neural Information Processing Systems, 33:15737–15749, 2020.
> > > >
> > > > [2] Yang Chen, Libo Zhang, Jiamou Liu, and Shuyue Hu. Individual-level inverse reinforcement learning for mean field games. In Proceedings of the 21st International Conference on Autonomous Agents and Multiagent Systems, pages 253–262, 2022.
> > > >
> > > > [3] Yang Chen, Libo Zhang, Jiamou Liu, and Michael Witbrock. Adversarial inverse reinforcement learning for mean field games. arXiv preprint arXiv:2104.14654, 2021.
> > > >
> > > > [4] Jiachen Yang, Xiaojing Ye, Rakshit Trivedi, Huan Xu, and Hongyuan Zha. Learning deep mean field games for modeling large population behavior. arXiv preprint arXiv:1711.03156, 2017.

---

### Official Review · Reviewer_MgoB · 2023-07-06

**Soundness:** 3 good
**Presentation:** 3 good
**Contribution:** 3 good
**Rating:** 6
**Confidence:** 3

**Summary:**

The authors prove new upper bounds on the error in imitation learning for mean-field games. They then propose a new approach based on using mean-field control, which might provide a more reasonable imitation goal for imitation learning.

**Strengths:**

The paper is clear in its description of connections and provides extensive mathematical arguments. The authors clearly show a gap in current literature and proceed with an ample analysis of how to fill it, which -- albeit very preliminary -- may lay substantial ground work for future research, as described in the paper's very own conclusion.

**Weaknesses:**

In my opinion, the paper's biggest weakness is that it talks about practical applications only in the introduction and then never again. The whole paper would benefit from presenting concrete examples for the various settings that are discussed and including references, where these settings might occur "in the wild". Most notably the link to practice is lacking in l.s 235-239 and 321-324. Numerical analysis was suggested in the paper and is, in fact, presented in the appendix to some extent, but might have also deserved a little bit of space in the main paper. As it is, it is hard to see the practical applications of the proposed techniques.

While related work is discussed very well, I do not see why it is pointed out as a separate contribution. Overall, the list of contributions could be stream-lined when strengthening the overall story (by referring to the beginning in the conclusion, e.g.).

The writing style is very good. Minor mistakes persist:
- l. 87 "policy, that" --> "policy that"
- l. 135: \pi_E ---> \pi^E
- l. 319: "kind" ---> "kinds"
- l. 379: "works open" --> "work opens"
- throughout: comma after "i.e." is set inconsistently.

**Questions:**

Which further studies should be executed to solidify empirical evidence and why did you choose the setting you did in the appendix? Can a  small explanation be fit into the main text?

**Limitations:**

The authors discuss the limitations of their work generally very well.

---

> ### Author Rebuttal · Authors · 2023-08-08
>
> We would like to thank the reviewer for the insightful comments. We would like to thank the reviewer also for recognizing the clarity of our work and how our paper  ¨shows a gap in current literature and proceeds with an ample analysis of how to fill it¨.
> We answer the raised questions below. Thank you for pointing out the typos, we corrected them according to your suggestions.
>
> ### Weaknesses
>
> - Practical applications are the main reason which drives our study on Imitation Learning for MFGs. Many real-world sequential decision making problems can be formulated as a MFG, from traffic routing to financial management or online business with a large customer body. We will extend the discussion about the relevant applications in the main paper. Furthermore, we will add the empirical evaluation in the main body, using the new results provided in the pdf file. It is also important to notice that, with our last experiment, we showed empirically the exponential dependency of BC and vanilla-ADV on the horizon.
>
> - We described the related work as one of the main contributions of the paper since with this paper we tried to harmonize the different solution concepts presented in the SotA. However, we have no objection in toning this down.
>
> ### Questions
>
> - The numerical simulation provides an experimental verification of the theoretical results presented in the paper. This hard-instance MFG is inspired by the hard-instance MDP used to show the lower bound for vanilla imitation learning. We provide a new experiment in the pdf file which shows empirically the exponential dependency of BC and vanilla-ADV on the horizon, for this hard-instance MFG.

---

### Official Review · Reviewer_4ip4 · 2023-07-07

**Soundness:** 2 fair
**Presentation:** 3 good
**Contribution:** 2 fair
**Rating:** 5
**Confidence:** 3

**Summary:**

In this work, the authors propose a newly defined measure called the Nash imitation gap for imitation learning in mean-field games (MFG). They establish upper bounds for the Nash imitation gap of behavioral cloning and adversarial imitation learning in both population independent and dependent dynamics in MFG. Additionally, they introduce the formulation of mean-field control (MFC) problem as a proxy to address population-dependent dynamics in imitation learning scenarios.

**Strengths:**

1.The authors of this paper demonstrate a high level of clarity in their definitions and statements, making the paper easily readable and accessible to the readers. This clarity enhances the understanding and comprehension of the presented concepts and results.

2.One notable contribution of this paper is the introduction of an alternative optimization goal for imitation learning in mean-field games (MFG). By adopting this alternative goal, the paper avoids the need for approximating the MFG dynamics driven by the expert population. This alternative approach provides a novel perspective and potential solution for tackling the challenges associated with MFG-driven imitation learning.

**Weaknesses:**

1.While the authors provide a numerical illustration in the appendx, the lack of comprehensive experiments weakens the justification for using the Nash imitation gap as a measure for quantifying the quality of imitation. The absence of quantitative evaluations and comparisons with other approaches limits the ability to fully assess the effectiveness and practical significance of the proposed measure.

2.Despite introducing a new perspective through the Nash imitation gap, this paper does not present any new algorithms or methods specifically designed for addressing this measure. This limitation reduces the overall significance and contribution of the work, as it does not offer novel practical solutions or techniques that directly leverage the Nash imitation gap.

3.The lack of clear motivation for the use of the newly proposed performance measure undermines the justification and relevance of its inclusion. Providing a comprehensive rationale would strengthen the understanding and significance of the contributions.

**Questions:**

Questions:

1.In line 169, could you provide a more concrete justification for the use of the L1 distance as a measure?

2.Line 238 mentions the need for the learned policy to be an equilibrium policy, but it's not clear why this is necessary. Could you elaborate on the significance and potential applications of requiring an equilibrium policy?

3.Could you justify the need for defining the Nash imitation gap when exploitability is already defined? Are these measures equivalent, and is it possible to simplify by using just one of them?

4.The distinction between the learner's population and expert's population in the paper reminds me of the difference between online imitation learning (e.g., DAgger) and offline imitation learning (e.g., behavior cloning). Are these pairs of concepts related, and if so, how do they connect to the ideas presented in the paper?

**Limitations:**

1.The lack of a clear motivation for the inclusion of the newly proposed performance measure undermines its justification and relevance.

2.The limited scope of experiments and the absence of quantitative evaluations and comparisons with other approaches weaken the justification for using the Nash imitation gap as a measure for quantifying the quality of imitation.

3.The ambiguous goal of mean-field games (MFG) and the using the Nash imitation gap as a performance measure may limit its applicability in real-world applications.

---

> ### Author Rebuttal · Authors · 2023-08-08
>
> We would like to thank the reviewer for the insightful comments and for recognizing the clarity and the novelty of the work. We answer the raised questions below.
>
> ### Weaknesses
>
> 1. ¨*the lack of comprehensive experiments weakens the justification for using the Nash imitation gap as a measure for quantifying the quality of imitation*¨. (answer also Limitations 2) In our work we provide the **first measure** to evaluate the performance of imitation learning algorithms for MFGs. We believe that providing a valid performance measure is a preliminary step needed to design a valid algorithm for a specific problem (in this case for imitation learning in MFGs). Moreover, our analysis suggests that with population-independent dynamics MFGs, single-agent imitation learning algorithms such as Behavioral Cloning or adversarial approaches will achieve good performances also in MFGs. This is a notable result, since previous works (add citations) are, from an abstract perspective, instances of what we called vanilla-ADV which shows similar performances of easier methods as BC (surprisingly not considered in the literature). To summarize, in the case of MFGs with population-independent dynamics, we can use single-agent IL algorithms whereas on MFGs with population-dependent dynamics, we need different algorithms than the ones used in literature. For the latter setting, we propose a new algorithm design principle (MFC-ADV), which leads to better performance.
>
>
> 2. ¨*this paper does not present any new algorithms or methods specifically designed for addressing this measure*¨. The main contribution of this paper is to provide a **systematic analysis** of the MFG imitation learning problem setting. Moreover, we showed which distance needs to be optimized, i.e., the MFC-ADV. We do not provide a practical algorithm to optimize this, but we explain which proxy should be used and considered. We believe that our work advances the study of the IL problem, providing a relevant measure of how good the imitation is (the NIG) and how to achieve good performances with respect to this metric (optimizing the MFC-ADV).
>
> 3. ¨*The lack of clear motivation for the use of the newly proposed performance measure*¨. (answer also Limitation 1-3) The single agent imitation learning problem aims at minimizing the imitation gap assuming the expert is optimizing an unknown reward function. The imitation gap is the value function difference between the policy we are recovering with the imitation learning algorithm and the expert’s policy, for the unknown reward function. In our work we proposed the Nash Imitation Gap, which is the **natural extension** of the imitation gap to the MFG setting. Doing this we are defining the problem setting, and the goal of imitation learning in MFGs. As pointed out by reviewer Qrta, our work ¨ proposes a novel solution concept, NIG, which generalized the classic imitation gap¨. We will add this explanation more clearly in the final version of the paper.
>
> ### Questions
>
> 1. ¨*justification for the use of the L1 distance as a measure*¨. The main reason to use the L1 distance is its simplicity for conveying the main message. However, the whole analysis can be easily extended to other distance metrics (as KL-divergence). In fact, using tools such as Pinsker inequality (as mentioned eg l.149-150); we would get similar bounds with $\sqrt{\epsilon}$ rather than $\epsilon$. This would not influence the main message of the paper which is the exponential dependency to the horizon of the BC and vanilla-ADV, whenever the dynamics does depend on the population.
>
> 2. ¨ *the need for the learned policy to be an equilibrium policy, but it's not clear why this is necessary.* ¨ For single-agent imitation learning is usually assumed that the observed policy is the optimal policy for some unknown reward function. We extended this concept to the MFG setting, assuming that the observed policy is a Nash equilibrium for some unknown reward function, and we quantified the quality of the imitation accordingly to it.
>
> 3. ¨ *justify the need for defining the Nash imitation gap when exploitability is already defined* ¨. In the single-agent imitation learning setting, the imitation gap is just the value function difference between the expert´s policy and the imitating policy. We extended this to the MFG setting. In fact, as stated in the paper (see l.242), "the NIG is simply defined as being the exploitability of the considered policy" (for the unknown reward function).
>
> 4. ¨ *Are these pairs of concepts related, and if so, how do they connect to the ideas presented in the paper?* ¨ Thanks for the interesting question. We can say yes and no. In fact, in the single-agent imitation learning we have only the policy as the state-action distribution induced by it. Then, in the offline and online setting are considered distance metrics which depends on the state-action distribution of the expert´s policy or the imitating policy. In the MFG setting we are on a different level, since we have both the policy and the population (which can be induced by a different policy). Then we need to consider if the population is induced by the imitating policy or by the expert´s one.

---

> > ### Comment · Reviewer_4ip4 · 2023-08-19
> > **Thanks for your explanations.**
> >
> > Dear Author,
> >
> > Thanks for addressing my questions and I am happy to increase my score to 5.
> >
> > Best regards,
> >
> > 4ip4

---

### Author Rebuttal · Authors · 2023-08-08

We would like to thank the reviewers for the time they spent reviewing our paper and for their valuable feedback.

All the reviewers recognized the **clarity** and the **novelty** of our work, providing **new** and **first** theoretical guarantees for this setting. Moreover, MgoB, underlined how our work ¨shows a gap in current literature and proceeds with an ample analysis of how to fill it¨; uQRX noticed that we ¨propose a new adversarial algorithm to mitigate the exponential dependency on the horizon¨; and Qrta wrote that our work  ¨suggests that the adversarial imitation learning could replace the inner RL by an MFC¨. Thanks for the positive feedback.

The main concerns raised relate to experimental evaluation (4ip4, MgoB, uQRX), and the used L1 distance (4ip4, uQRX).

- **Experimental evaluation.** The concern mainly regards the absence of it in the main body of the paper and the fact that this did not show the exponential dependency of BC and vanilla-ADV on the horizon. We provide in the pdf file attached to the rebuttal a new experiment which shows empirically the exponential dependency of BC and vanilla-ADV on the horizon. We will add the results of this experiment to the main text of the paper.

- **L1 distance.** The main reason to use the L1 distance is its simplicity for conveying the main message. However, the whole analysis can be easily extended to other distance metrics (as KL-divergence). In fact, using tools such as Pinsker inequality (as mentioned eg l.149-150); we would get similar bounds with $\sqrt{\epsilon}$ rather than $\epsilon$. This would not influence the main message of the paper, which is the exponential dependency to the horizon of the BC and vanilla-ADV, whenever the dynamics does depend on the population.

We hope our answers clarify the raised questions about our work. We will be happy to take any further questions they might have during the discussion period.

---

### Decision · Program_Chairs · 2023-09-21

**Decision:**

Accept (poster)

**Comment:**

This paper presents a novel solution concept, NIG, which generalized the classic imitation gap, and shows the upper-bounds of NIG for several different settings. Reviewers agree that the theoretical analysis is solid and supports the upper bounds well.  The paper is clearly written and well organized, and both the theoretical upper bounds and the empirical results suggest that the adversarial imitation learning could replace the inner RL by an MFC. There were some concerns about the gap between the theoretical results and empirical algorithm - the authors acknowledge those gaps but make the argument that the novel theoretical result is the main contribution, namely that there is an exponential gap between single agent and mean-field games in the imitation learning settings. I am voting to accept this paper mainly due to this contribution. I encourage the authors to be more clear about the empirical gap and add discussion about this and the ramifications in the camera ready version.